# Tumour-elicited neutrophils engage mitochondrial metabolism to circumvent nutrient limitations and maintain immune suppression

Christopher M. Rice[1], Luke C. Davies[1,2], Jeff J. Subleski[1], Nunziata Maio[3], Marieli Gonzalez-Cotto[1], Caroline Andrews[1], Nimit L. Patel[1], Erika M. Palmieri[1], Jonathan M. Weiss[1], Jung-Min Lee[4], Christina M. Annunziata[4], Tracey A. Rouault[3], Scott K. Durum[1] & Daniel W. McVicar[1]

Neutrophils are a vital component of immune protection, yet in cancer they may promote tumour progression, partly by generating reactive oxygen species (ROS) that disrupts lymphocyte functions. Metabolically, neutrophils are often discounted as purely glycolytic. Here we show that immature, c-Kit[+] neutrophils subsets can engage in oxidative mitochondrial metabolism. With limited glucose supply, oxidative neutrophils use mitochondrial fatty acid oxidation to support NADPH oxidase-dependent ROS production. In 4T1 tumour-bearing mice, mitochondrial fitness is enhanced in splenic neutrophils and is driven by c-Kit signalling. Concordantly, tumour-elicited oxidative neutrophils are able to maintain ROS production and T cell suppression when glucose utilisation is restricted. Consistent with these findings, peripheral blood neutrophils from patients with cancer also display increased immaturity, mitochondrial content and oxidative phosphorylation. Together, our data suggest that the glucose-restricted tumour microenvironment induces metabolically adapted, oxidative neutrophils to maintain local immune suppression.

[1] Cancer & Inflammation Program, National Cancer Institute, Frederick, MD 21702, USA. [2] Division of Infection & Immunity, School of Medicine, Cardiff University, Tenovus Building, Heath Park, Cardiff CF14 4XN, UK. [3] Molecular Medicine Branch, National Institute of Child Health and Human Development, National Institute of Health, Bethesda, MD 20892, USA. [4] Women's Malignancies Branch, Center for Cancer Research, National Cancer Institute, Bethesda, MD 20892, USA. Correspondence and requests for materials should be addressed to C.M.R. (email: chris28rice@gmail.com) or to D.W.M. (email: mcvicard@mail.nih.gov)

The importance of neutrophils in the establishment and progression of tumours has been widely reported, with multiple tumour modulatory functions identified. Neutrophils promote tumour progression by increasing invasion and metastasis through releasing proteases[1,2], increasing angiogenesis[3,4] and directly promoting tumour growth[5–7]. Additionally, neutrophils have been shown to limit anti-tumour immune responses by suppressing T cell[8–10] and NK-cell[11] activity, through various factors such as arginase[12] and reactive free radicals; namely reactive nitrogen[13] and oxygen[14] species. Neutrophils have also been included as part of the myeloid-derived suppressor cells (MDSC), a mixed population of myeloid cells, including immature neutrophils and monocytes, defined by their immune suppressive and tumour promoting capacity[9,15–17]. However, consensus on the separation between the granulocytic (g)MDSC subset and neutrophils is controversial and this nomenclature implies that neutrophils in the cancer setting are purely immunosuppressive and pro-tumour in nature, despite described anti-tumour functions[18]. This confusion can be attributed to the unknown predictive factors which lead to this neutrophil functional dichotomy.

In recent years, advances in our understanding of metabolism and its effects on cell function has opened new avenues in leukocyte biology, impacting both cancer biology and immunology. Tumours and immune cells engage in bi-directional manipulation of their respective metabolism, thereby altering cell function to facilitate tumour progression[19,20]. This can occur by direct reprogramming of target cell metabolism[21] or by competition for fuels in the tumour microenvironment (TME) leading to starvation or adaptions to metabolic stress[22]. However, despite our recent advances in the understanding of leukocyte metabolism and the known importance of neutrophils in tumour progression, neutrophil metabolic status and the impact it plays on effector functions remain poorly studied. Traditionally, neutrophils have been thought to be a highly glycolytic population, dependent on glucose, with little or no mitochondrial function except to drive apoptosis[23]. Recently however, neutrophil metabolism has gained interest as the importance of mitochondria in effector functions such as chemotaxis[24] and the generation of neutrophil extracellular traps (NETs) have come to light[25]. Recently, MDSC function was shown to have specific metabolic underpinnings, with tumour-associated MDSC exhibiting increased fatty acid oxidation and mitochondrial function that correlated with immunosuppressive capacity[26]. These studies however, did not define the MDSC sub-populations, dissect how mitochondrial function was enhanced or delineate how it contributed to the suppressive action of these cells. Therefore, it is possible, that a role for neutrophil metabolic programming has overlooked due to their inclusion within the broad category of MDSC.

In this study, we characterise neutrophil mitochondrial function under healthy physiological conditions and in the cancer setting. We demonstrate that tumour elicited c-Kit signalling in neutrophils drives an oxidative phenotype with enhanced mitochondrial function. Surprisingly, we find that oxidative neutrophils utilize their mitochondria to maintain intracellular NADPH levels and support reactive oxygen species (ROS) production through NADPH oxidase when glucose utilisation is limited. Our data suggest that oxidative neutrophils benefit tumour growth as, unlike glycolytic-neutrophils from a healthy host, they can maintain ROS-mediated suppression of T cells even in the nutrient limited TME. These data shed new light on the role of mitochondria in phagocyte function and suggest that neutrophil mitochondrial metabolism could prove an effective target for cancer therapy.

## Results

**c-Kit is a marker of immature oxidative neutrophils.** Very little is known about neutrophil mitochondrial activity and how it may relate to their developmental state or function in different tissue settings. Therefore, initial experiments characterised neutrophil mitochondrial function in distinct tissue locations. Bone marrow-derived neutrophils possessed significantly greater basal mitochondrial oxygen consumption rate (OCR) when compared to neutrophils from peripheral blood or spleen (Fig. 1a). As bone marrow is the primary site of haematopoiesis, we hypothesised that elevated mitochondrial capacity might be related to an immature phenotype. Expression of the chemokine receptor CXCR2, and the stem cell factor (SCF) receptor c-Kit, markers of neutrophil maturity and immaturity respectively[27], identified two distinct populations of Ly-6G+ cells (Fig. 1b, Supplemental Fig. 1a). Nuclear morphology confirmed that c-Kit+/CXCR2− neutrophils were a more immature subset compared to c-Kit−/CXCR2+ cells, with a higher percentage of unsegmented nuclei with an immature chromatin pattern (Fig. 1c and Supplemental Fig. 1b). c-Kit+ neutrophils did not express monocyte markers Ly-6C or CD115 (MCSFR) (Supplemental Fig. 1c), but instead expressed markers consistent with mature c-Kit− neutrophils such as CD62L and CD11b (Supplemental Fig. 1d). Accordingly, c-Kit expression in neutrophils was largely confined to the bone marrow with low percentages found in the spleen and circulation (Fig. 1d). Following isolation (Supplemental Fig. 1e), seahorse extracellular flux analysis demonstrated that c-Kit+ neutrophils possess a significantly higher reserve and maximal mitochondrial OCR than c-Kit− neutrophils (Fig. 1e). Moreover, analysis of extra-cellular acidification rates (ECAR), a measurement related to the production of lactic acid and therefore the glycolytic rate, revealed that OCR/ECAR ratios of c-Kit+ neutrophils were substantially higher than c-Kit− cells (Fig. 1f), suggesting a greater propensity for mitochondrial function as opposed to glycolysis to meet energy demands. Further investigation identified a bone marrow resident c-Kit+ ly6Gint population (Supplemental Fig. 2a) with a greater reserve and maximal mitochondrial OCR (Supplemental Fig. 2b,c). This population displayed a more myelocyte-like nuclear morphology[9] (Supplemental Fig. 2d), a reduced expression of Ly-6G and increased expression of Ly-6C and MCSF-R (CD115) (Supplemental Fig. 2e,f), suggesting that this population represents granulocyte/monocyte myelocyte precursor and that loss of mitochondrial metabolism may be indicative of neutrophil development.

Consistent with a greater capacity for oxidative phosphorylation (OXPHOS), tetramethylrhodamine, ethyl ester (TMRE) staining determined that c-Kit+ neutrophils possess greater mitochondrial membrane potential when compared to c-Kit− neutrophils (Fig. 1g). Additionally, c-Kit+ neutrophil mitochondria possess greater quantities of mitochondrial complexes I, II, III and V (Fig. 1h), mitochondrial enzymes pyruvate dehydrogenase (PDH), α-ketoglutarate dehydrogenase (αKGDH) and aconitase 2 (ACO2) (Supplemental Fig. 2g). Furthermore, these cells contained increased mitochondrial DNA (Supplemental Fig. 2h) consistent with increased mitochondrial mass. In-gel activity assays also showed greater activity for mitochondrial complex I, II and IV (Supplemental Fig. 2i).

c-Kit+ neutrophils displayed a distinct metabolite profile compared to c-Kit− neutrophils (Supplemental Fig. 2j). Prominently, arginine metabolism, which can support tissue repair and proliferation[28], was identified by ingenuity pathway analysis (IPA) of metabolite content, with higher ornithine and subsequently spermidine and proline in c-Kit+ neutrophils. In addition, the levels of nine amino acids were increased in c-Kit+ neutrophils, which may indicate higher levels of

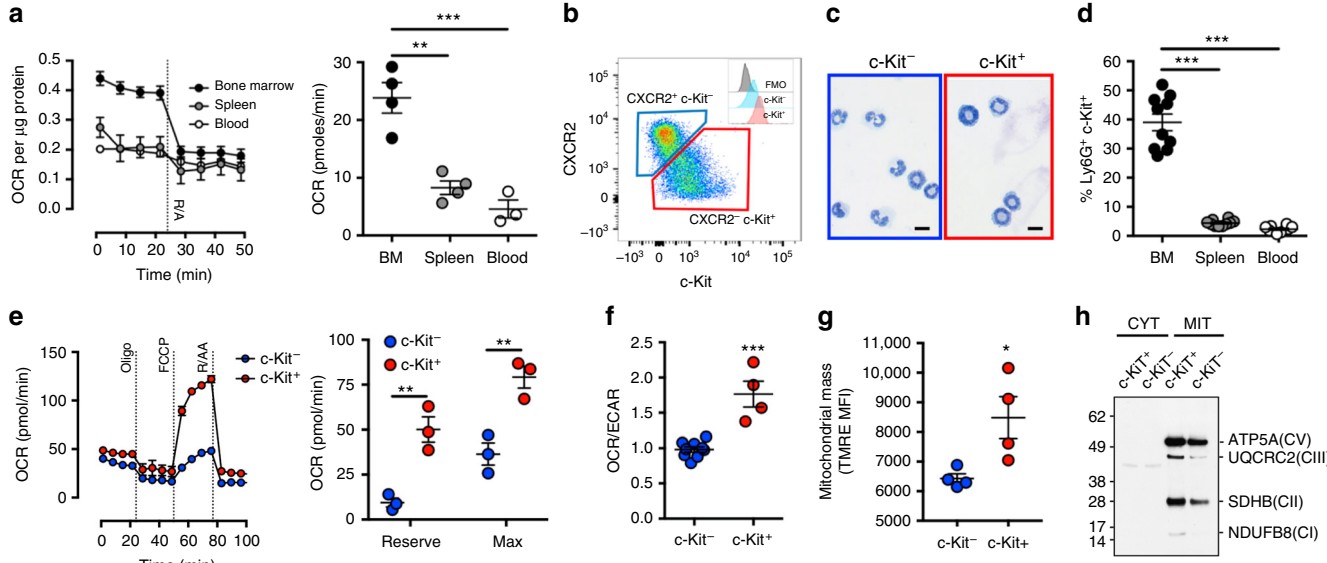

**Fig. 1** c-Kit is a marker of immature oxidative neutrophils. **a** Determination of C57BL/6J neutrophil basal mitochondrial oxygen consumption rates (OCR, left) following inhibition of electron transport chain components (Rotenone 100 nM and antimycin A 1 μM). Quantification of basal mitochondrial OCR from multiple independent experiments (right). Data were analysed by one-way ANOVA with Tukey's post-tests indicated. Bone marrow and spleen $n = 4$, blood $n = 3$. **b** Representative flow cytometry plot of Ly6G$^+$ neutrophils displaying c-Kit$^+$ CXCR2$^-$ and c-Kit$^-$ CXCR2$^+$ populations from naive C57BL/6 bone marrow. Histograms represent c-Kit expression of populations. **c** Representative nuclear morphology of bone marrow neutrophils following cytospin and Romanowsky staining. Scale bar represents 10 μm. **d** Percentage of neutrophils expressing c-Kit isolated from different tissues (BM—bone marrow). Data ($n = 10$ per group) represents two independent experiments, data were analysed by one way ANOVA, Tukey's post-tests are indicated on the graph. **e** Representative mitochondrial stress test (left) in bone marrow neutrophils (oligomycin 1.26 μM, Carbonyl cyanide-4-(trifluoromethoxy) phenylhydrazone (FCCP) 660 nM, Rotenone 100 nM and antimycin A 1 μM). Mitochondrial reserve and maximal OCR are quantified from multiple independent experiments (right). Data ($n = 3$ per group) were analysed by two-way ANOVA with Tukey's post-tests indicated. **f** OCR:extracellular acidification rate (ECAR) ratios of neutrophil subsets. Data (c-Kit$^+$ $n = 3$, c-Kit$^-$ $n = 9$) were analysed by Student's $t$-test and is representative of three independent experiments. **g** Median fluorescent intensity of tetramethylrhodamine ethyl ester (TMRE) staining, which indicates active mitochondrial content in neutrophil subsets. Data were analysed by Student's $t$-test, data ($n = 4$ per group) represents two independent experiments. **h** Western blotting for mitochondrial complexes I (NDUFB8), II (SDHB), III (UQCRC2) and V (ATP5A) in neutrophil subsets. CYT cytosolic, MIT mitochondrial. Data is representative of two repeats ($n = 3$). $p$ Values *<0.05, **<0.01, ***<0.001. All error bars represent the mean ± SEM

anaplerotic metabolism[29]. Collectively these data suggest that c-Kit expression is a marker of an immature neutrophil subset, largely restricted to the bone marrow with a drastically different metabolic programming compared to their more mature c-Kit$^-$ counterparts, as evidenced by their metabolic profile and higher mitochondrial capacity.

**Neutrophil mitochondria facilitate free radical production.** Neutrophils engage in the generation of ROS and other free radicals in a process termed respiratory burst. This process was originally characterised as an increase in oxygen consumption as molecular oxygen is consumed to form free radicals such as superoxide ($O_2^-$) and hydrogen peroxide ($H_2O_2$). Therefore, we dissected the metabolic underpinning of respiratory burst by monitoring the substantial increase in OCR in response to the PKC agonist phorbol 12-myristate 13-acetate (PMA) in bone marrow-derived neutrophils. Inhibition of mitochondrial function with rotenone and antimycin A (complex I and III inhibitors, respectively) during respiratory burst did not affect peak OCR but rather lead to reductions during the later phases of the response (Fig. 2a, Supplemental Fig. 3a). In contrast, the contribution of glucose utilisation was assessed using the competitive inhibitor 2-deoxy-D-glucose (2DG) to investigate glucose-independent pathways. 2DG treatment altered both the peak and the kinetics of respiratory burst, reducing OCR in the initial earlier phase while the later phase remained unaffected (Fig. 2a, Supplemental Fig. 3a).

Unexpectedly for a reportedly glucose-dependent cell, there remained substantial OCR following the disruption of glucose metabolism. To uncover the metabolic components of respiratory burst during 2DG restriction of glucose metabolism, we employed mitochondrial inhibition following PMA stimulation. Mitochondrial inhibition led to dramatic reductions in OCR during 2DG treatment, demonstrating mitochondrial involvement during glucose-independent respiratory burst (Fig. 2b, Supplemental Fig. 3b). Conversely, inhibition of glucose utilisation after mitochondrial complex inhibition also significantly limited respiratory burst (Fig. 2b, Supplemental Fig. 3b). These data strongly suggested that bone marrow neutrophils maintain respiratory burst via both glucose metabolism and mitochondrial function. Although OCR is indicative of respiratory burst, it is not a direct measurement of ROS production. Therefore, $H_2O_2$ production was directly measured in PMA activated neutrophils following disruption of glucose utilisation or mitochondrial function (Fig. 2c). Consistent with our OCR data, blockade of mitochondrial function alone during respiratory burst did not affect peak ROS production but inhibited late $H_2O_2$ production (Fig. 2c). In contrast, blockade of glucose utilisation by 2DG revealed a distinct ROS production profile, reduced in amplitude and prolonged in length when compared to mitochondrial inhibition. Treatment with acetylcarnitine transport inhibitor etomoxir, showed that fatty acid utilisation is required to maintain PMA-induced OCR (Fig. 2d, Supplemental Fig. 3c) following treatment with 2DG and both fatty acid metabolism and mitochondrial function was required $H_2O_2$ production under

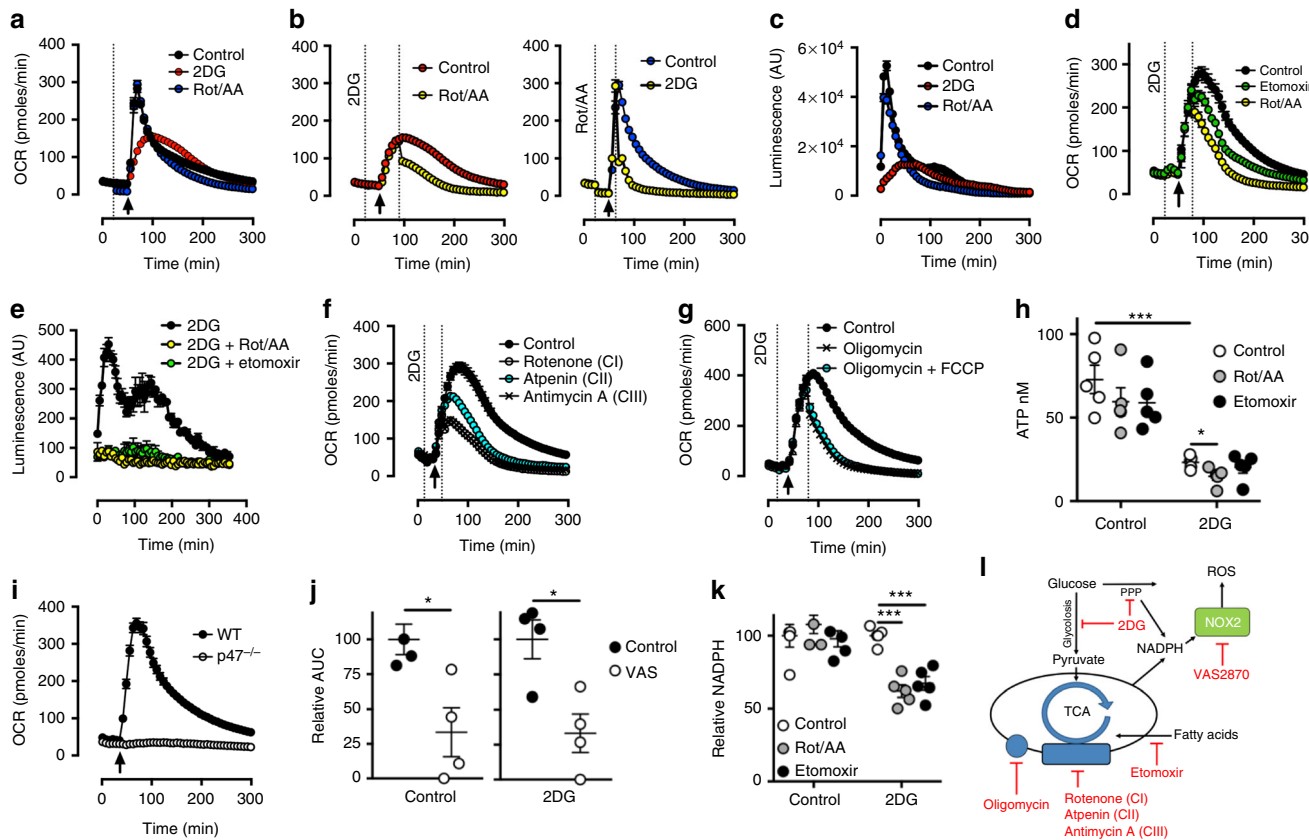

**Fig. 2** Neutrophil mitochondria facilitate free radical production. **a** C57Bl/6J bone marrow neutrophil respiratory burst was measured by oxygen consumption rate (OCR) in response to phorbol 12-myristate 13-acetate (PMA) at the arrow following the addition of indicated compounds at the dotted line. **b** Following stimulation as described in panel (**a**), neutrophils received a second stimulation of indicated compounds at the second line. Control from panel (**a**). **c** $H_2O_2$ production from bone marrow neutrophils in response to PMA 20 min after the addition of the indicated individual compounds. **d** OCR in response to the indicated compounds at the second line following the addition of PMA (arrow) and 2DG (first line). **e** $H_2O_2$ production in response to PMA 20 min after addition of 2DG alone or in combination with the indicated compounds. **f** OCR values in response to Inhibitors to electron transport chain complexes (CI-III) added at the second line following stimulation with PMA (arrow) and 2DG (first line). **g** OCR in response to the complex V inhibitor Oligomycin and Carbonyl cyanide-4-(trifluoromethoxy)phenylhydrazone (FCCP) were added at the second line, following stimulation with PMA (arrow) and 2DG (first line). **h** ATP content of PMA-stimulated bone marrow neutrophils following inhibition with indicated compounds in the presence or absence of 2DG. Data ($n = 5$ per group) were analysed by two-way ANOVA with Tukey's multiple comparisons indicated. **i** OCR in $p47^{-/-}$ neutrophils in response to PMA (arrow). **j** Relative total oxygen consumption in response to PMA during NADPH oxidase inhibition by VAS-2870 in the presence or absence of 2DG. Data were analysed by Student's $t$-test. Control $n = 3$ pooled from three independent experiments, 2DG $n = 4$ pooled from two independent experiments. **k** Relative NADPH levels in PMA-stimulated neutrophils in the presence or absence of 2DG following treatment with indicated compounds. Data ($n = 5$) representative of two independent experiments and were analysed by two-way ANOVA with Tukey's multiple comparisons indicated. **l** Schematic displaying our metabolic model of ROS support, with indicated inhibitors. Concentrations were: Rotenone (Rot, 100 nM), Atpenin (1 μM), Antimycin A (AA, 1 μM), 2-DG (100 mM), PMA (1 μg/ml), FCCP (660 nM), etomoxir (100 μM), Oligomycin (1.26 μM), and VAS-2870 (10 μM). $p$ Values *<0.05, ***<0.001. All error bars denote the mean ± SEM

these conditions (Fig. 2e). Together these data show that optimal ROS production requires two distinct metabolic pathways, with glucose metabolism being required for early phase high-intensity ROS production and mitochondrial function facilitating late phase prolonged $H_2O_2$ production.

Recent studies have demonstrated that respiratory burst in macrophages is dependent on mitochondrial function and involves increased dependency on mitochondrial complex II and the decoupling of mitochondria from OXPHOS by complex V[30,31]. To assess the involvement of individual mitochondrial complexes during glucose-independent respiratory burst in neutrophil, we used specific inhibitors to components of the mitochondrial electron transport chain (ETC). Inclusion of inhibitors to mitochondrial complex I, II or III all significantly reduced glucose-independent respiratory burst (Fig. 2f, Supplemental Fig. 3d). Contrary to previous reports in macrophages, complex V (ATP synthase) inhibition also ablated glucose-

independent respiratory burst in neutrophils (Fig. 2g, Supplemental Fig. 3e). Inhibition of complex V not only inhibits ATP production, but also suppresses electron flow through the ETC by blocking the recycling of hydrogen ions into the mitochondrial matrix. To differentiate between these possibilities, we used the mitochondrial uncoupler carbonyl cyanide-4(trifluoromethoxy) phenylhydrazone (FCCP) alongside oligomycin to allow hydrogen recycling and electron transport to the matrix in the absence of ATP synthesis. These data demonstrated that in neutrophils ATP production by complex V and not simply proton recycling was required for respiratory burst (Fig. 2g, Supplemental Fig. 3e). Direct measurement of ATP showed that the majority of ATP was maintained by glucose metabolism, however in the absence of glucose utilisation, mitochondrial function was required to maintain ATP levels (Fig. 2h). Conversely, disruption of fatty acid metabolism which suppressed respiratory burst (Fig. 2d, e) did not significantly reduce ATP levels, suggesting that

maintenance of ATP production alone may not fully account for the mechanism by which neutrophils engage in respiratory burst in the absence of glucose metabolism.

Respiratory burst in neutrophils is heavily dependent on the NADPH-oxidase (NOX) complex which utilises NADPH to generate superoxide, yielding $NADP^+$. However previous studies have demonstrated that mitochondria can contribute to respiratory burst in macrophages and neutrophils by acting as a direct generator of ROS[30,32]. It is therefore possible that in the absence of glucose utilisation, bone marrow-derived oxidative-neutrophils could switch their source of ROS generation from NOX to mitochondria. PMA stimulated neutrophils deficient in the NOX2 subunit p47 ($p47^{-/-}$) were unable to undergo a significant respiratory burst (Fig. 2i). However, ROS production was detected by the fluorescent probe 2′,7′-dichlorofluorescin diacetate (DCFDA) in $p47^{-/-}$ neutrophils following PMA stimulation, albeit to a lesser extent than wild type neutrophils and was not detected by aminophenyl fluorescein (APF) (Supplemental Fig. 3f). This activity coincided with an increase in rotenone and antimycin A sensitive OCR following exposure to PMA in $p47^{-/-}$ neutrophils, suggesting that that the ROS detected by DCFDA was due to an increase in mitochondrial activity (Supplemental Fig. 3g). This activity was further increased following 2DG treatment, suggesting that increased mitochondrial activity may be an adaption to maintain responses when glucose metabolism is limited (Supplemental Fig. 3g).

Furthermore, using the NOX inhibitor VAS2870, PMA-induced respiratory burst was found to be equally dependent on NOX activity in the presence and absence of intact glucose metabolism (Fig. 2j). These data suggest that in the absence of glucose, mitochondria are not functioning as the site of ROS production, as has been reported in macrophages[30], but may support NOX activity through NADPH production. To investigate this possibility, we measured NADPH levels during PMA stimulation (Fig. 2k). These data demonstrated that inhibition of mitochondrial function or fatty acid usage alone had no effect on NADPH maintenance. Remarkably however, when glucose usage is limited the inhibition of either mitochondrial function or fatty acid metabolism significantly reduced NADPH levels (Fig. 2k).

Taken together, these data suggest that neutrophils are able to utilize two metabolic pathways to maintain respiratory burst activity. Initially ROS production is dependent on glucose utilisation to maintain cellular ATP and NADPH levels. However, when glucose metabolism is limiting, neutrophils become dependent on mitochondrial metabolism to support cellular ATP and in particular fatty acid metabolism to support NOX activity, through the maintenance of NADPH levels (Fig. 2l).

**Mitochondrial capacity aids ROS production in low glycolysis.** Having established that mitochondrial function can support respiratory burst when glucose usage has been inhibited, we hypothesized that neutrophil with greater mitochondrial function may better maintain ROS production in the absence of glucose. To address this possibility, we compared neutrophils from the bone marrow and spleen, two populations with differing mitochondrial capacity (Fig. 1a). Splenic neutrophils were unable to maintain respiratory burst at lower concentrations of glucose or treatment with 2DG (Fig. 3a). In contrast, bone marrow-derived neutrophils maintained respiratory burst in low glucose or during 2DG treatment. Moreover, c-Kit$^+$ neutrophils were better able to maintain respiratory burst in limiting glucose when compared to c-Kit$^-$ neutrophils (Fig. 3b). c-Kit$^+$ neutrophils also displayed sensitivity to mitochondrial complex inhibitors when undergoing respiratory burst in response to the toll-like receptor (TLR) agonist zymosan (Supplemental Fig. 4a), suggesting that mitochondrial function may also play a role in response to pathogens in immature neutrophils. Together these data demonstrate that neutrophils with enhanced mitochondrial capacity, can more effectively maintain respiratory burst when glucose utilisation is restricted.

**4T1 tumours elicit mitochondrial metabolism in neutrophils.** Neutrophils are often expanded in tumour bearing mice and cancer patients. Tumour elicited neutrophils have previously been characterised as a type of MDSC, a heterogeneous population of immature neutrophils and monocytes with functional differences from healthy blood neutrophils. Therefore, we asked whether the immature neutrophils expanded in tumour bearing mice might also possess enhanced mitochondrial function.

4T1 mammary tumours produce the c-Kit ligand, SCF[33], and have been shown to elicit expansion of Ly6G$^{hi}$ Ly6C$^{low}$ c-Kit$^+$ myeloid cells[34]. Indeed, neutrophils were significantly expanded in the bone marrow, spleen and blood (Supplemental Fig. 5a) of mice bearing 4T1 tumours, with large numbers infiltrating the TME (Supplemental Fig. 5b). Moreover, we found that both the frequency of c-Kit$^+$ neutrophils and the surface expression of c-Kit on these cells was significantly increased in the circulation of tumour bearing mice (Fig. 4a and Supplemental Fig. 5c). However, expression of Ly-6G, CD62L and CXCR2 were not

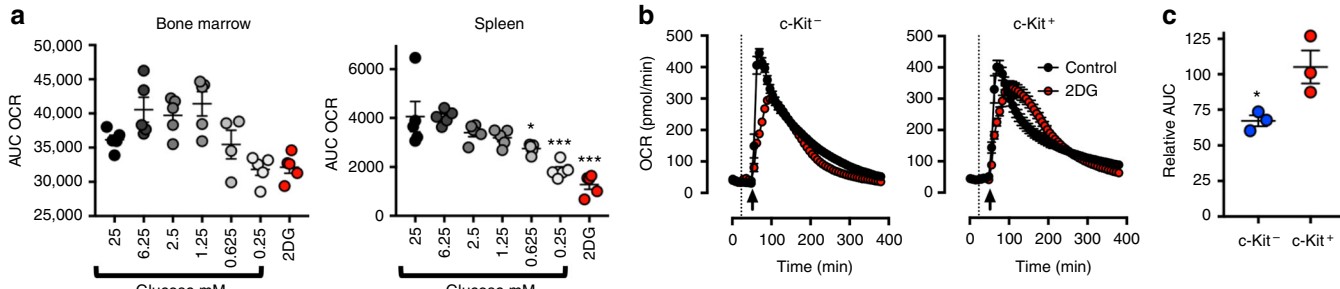

**Fig. 3** Mitochondrial capacity aids ROS production in low glycolysis. **a** Area under the curve (AUC) oxygen consumption rate (OCR) (total oxygen consumption in pmol) of respiratory burst in bone marrow- and spleen-derived C57BL/6J neutrophils following stimulation with phorbol 12-myristate 13-acetate (PMA, 1 μg/ml) in 0.25–25 mM glucose or following 2-deoxy glucose (2DG, 100 mM). Data ($n = 5$ per group) were analysed by paired one-way ANOVA with Dunnett's multiple comparison tests indicated, data represents two similar experiments. **b** Representative OCR traces from C57BL/6 bone marrow neutrophil subsets following stimulation with PMA (1 μg/ml) (arrow) in the presence or absence of 2DG (100 mM) (dotted line). **c** Relative AUC OCR in neutrophil subsets from (**b**) following 2DG and PMA stimulation, when compared to PMA alone. Data ($n = 3$ per group) were analysed by paired t-test, from two independent experiments. p Values *<0.05, ***<0.001. All error bars display mean ± SEM

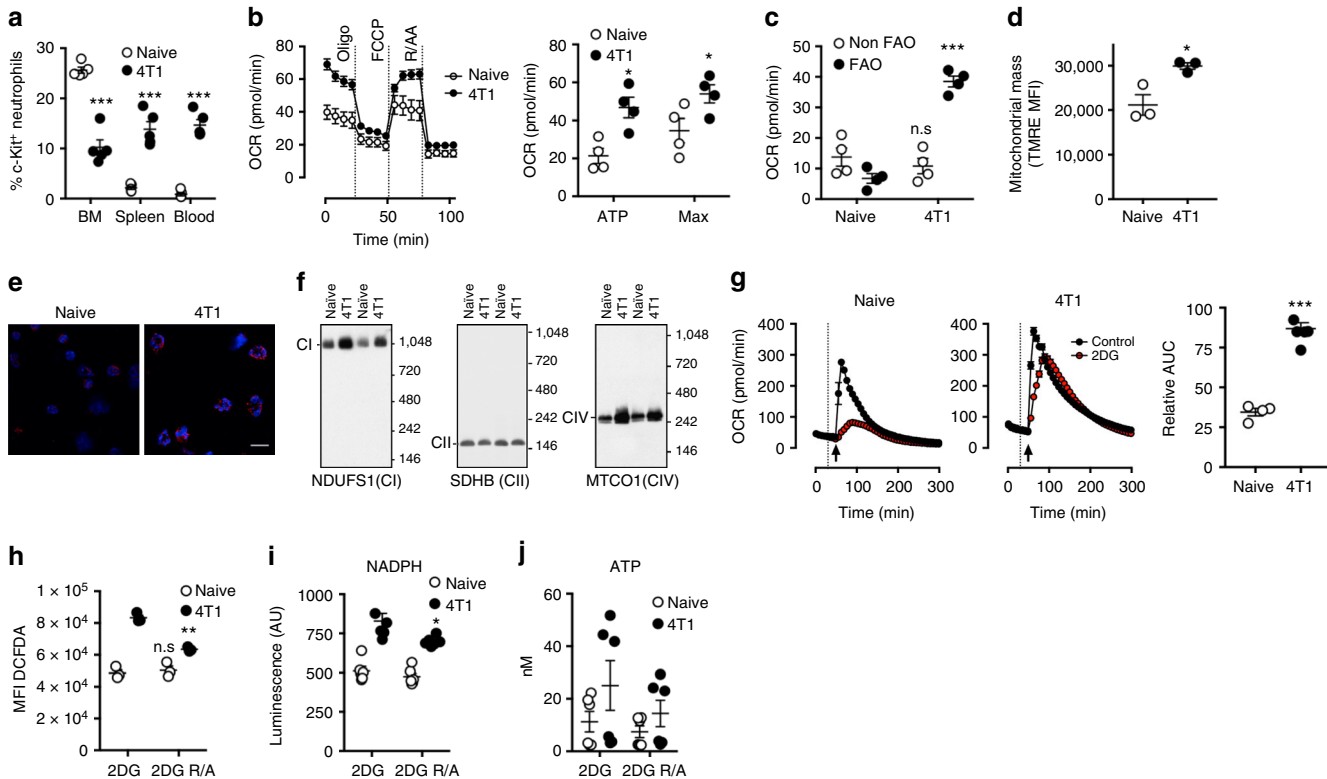

**Fig. 4** 4T1 tumours elicit mitochondrial metabolism in neutrophils. **a** The percentage of neutrophils that are c-Kit+ in naive and 4T1-bearing Balb-c mouse tissues ($n = 5$ per group). **b** Representative mitochondrial stress tests (left) of splenic neutrophils from naive and 4T1-bearing mice (oligomycin (Oligo), Carbonylcyanide-4(trifluoromethoxy)phenylhydrazone (FCCP), rotenone (R) and antimycin A (AA). ATP synthase-dependent and maximal oxygen consumption rate (OCR) quantified from multiple experiments (right, $n = 4$ per group). **c** ATP synthase OCR that is dependent on fatty acid oxidation (FAO) was assessed using etomoxir. Data ($n = 4$ per group) from two independent experiments. **d** Median fluorescent intensity (MFI) of tetramethylrhodamine ethyl ester (TMRE) staining in splenic neutrophils from naive and tumour-bearing mice ($n = 3$). **e** Representative images of splenic neutrophils from tumour-bearing and naive mice stained with mitotracker CMXROS (100 μM) and Hoechst 33342 (3.25 μM), scale bar represents 20 μm. **f** Native immunoblots for mitochondrial complex I (CI, NDUFS1) Complex II (CII, SDHB) and Complex IV (CIV, MTCO1) of splenic neutrophils from naive and 4T1-bearing mice. **g** Representative OCR of splenic neutrophils from tumour-bearing or naive mice following stimulation with phorbol 12-myristate 13-acetate (PMA) (arrow) in the presence or absence of 2-deoxy glucose (2DG) (dotted line). Quantification of area under the curve (AUC) of 2DG-stimulated neutrophils relative to PMA alone from multiple experiments (naive $n = 4$, 4T1 $n = 6$). **h** MFI of 2′,7′-dichlorofluorescin diacetate (DCFDA) (5 μM) in splenic neutrophils from tumour-bearing or naive mice in response to PMA (20 min) following stimulation with the indicated compounds ($n = 3$). **i, j** NADPH and ATP levels in splenic neutrophils from tumour-bearing or naive mice following stimulation with PMA in the presence the indicated compounds. Data ($n = 6$) are from two independent experiments. Data were analysed by two-way ANOVA with Sidak's (**h–j**) or Tukey's (**a–c**) multiple comparisons indicated, or Student's t-test (**d, g**). All error bars show mean ± SEM. Concentrations were: Rotenone (100 nM), Antimycin A (1 μM), 2-DG (100 mM), PMA (1 μg/ml), FCCP (660 nM), Oligomycin (1.26 μM), VAS-2870 (10 μM) and etomoxir (100 μM). p Values *<0.05, **<0.01, ***<0.001. All error bars display mean ± SEM

altered in 4T1 elicited neutrophils (Supplemental Fig. 5d) and immature nuclear morphology did not associate with c-Kit expression (Supplemental Fig. 5e). Assessment of mitochondrial metabolism demonstrated that 4T1 elicited splenic neutrophils possessed significantly increased ATP synthase dependent and maximal OCR (Fig. 4b) attributable to an increase in fatty acid oxidation (Fig. 4c). Increased mitochondrial metabolism was also observed in blood circulating neutrophils from 4T1 bearing mice, suggesting that this metabolic programming was systemic (Supplemental Fig. 5f). Assessment of tumour associated neutrophil metabolism proved technically challenging, with ex vivo measurement of ECAR or OCR proving unfruitful (Supplemental Fig. 5g) and as such splenic neutrophils are investigated as a 4T1 elicited population.

4T1 elicited neutrophils possessed a greater mitochondrial membrane potential (Fig. 4d) and imaging of active mitochondria in 4T1 elicited and naive splenic neutrophils supported this (Fig. 4e). Assessment of fully-assembled respiratory chain complexes by native immunoblot revealed that neutrophils from tumour bearing mice had increased levels of complexes I

(NDUFS1) and IV (MTCO1), and significantly increased complex IV activity by in gel activity assay (Fig. 4f, Supplemental Fig. 5h). Interestingly, unlike bone-marrow derived c-Kit+ neutrophils from naive mice, mitochondrial complexes II (SDHB) (Fig. 4f) and III (UQCRFS1) (Supplemental Fig. 5i) were not increased in 4T1 elicited neutrophils. Furthermore, both naive and 4T1 elicited neutrophils displayed similar complex II activity (Supplemental Fig. 5h). This suggests that mitochondrial capacity in tumour elicited neutrophils, although similarly enhanced, is distinct from immature neutrophils in naive mice and perhaps this capacity is differentially utilised. Analysis of metabolite content revealed that, similar to c-Kit+ bone marrow neutrophils, 4T1 elicited neutrophils had higher levels of multiple amino acids as compared to naive neutrophils, again suggesting higher levels of anaplerotic metabolism[29]. Furthermore, IPA highlighted enrichment of the glutathione synthesis pathway in 4T1 elicited neutrophils, which is indicative of enhanced antioxidant capacity[35] (Supplemental Fig. 5i) and perhaps a response to increased ROS production associated with enhanced mitochondrial function.

4T1 elicited neutrophils were better able to maintain respiratory burst when glucose metabolism was limited, when compared to naive neutrophils (Fig. 4g). OCR measurements of respiratory burst was supported by DCFDA staining (Fig. 4h). Here, enhanced glucose-independent ROS of 4T1 elicited neutrophils was sensitive to mitochondrial inhibition. Lastly, similar to c-Kit$^+$ cells from bone marrow, tumour-elicited neutrophils maintained increased cellular NADPH levels through mitochondrial function during inhibition of glucose utilisation (Fig. 4i). ATP levels followed a similar pattern but were not significantly affected (Fig. 4i). Together these data suggest that enhanced c-Kit expression in 4T1 bearing mice correlates with an altered metabolic state in peripheral neutrophil populations where mitochondrial metabolism is increased due to increases in fatty acid utilisation required for maintenance of NADPH levels.

**SCF-c-Kit signalling drives neutrophil metabolic adaptations.** Next, we investigated how 4T1 tumours support the altered

metabolic phenotype in neutrophils. The correlation between c-Kit expression and mitochondrial metabolism in bone marrow neutrophils (Fig. 1e, g, h) and 4T1 elicited neutrophils (Fig. 4a, b), together with the production of Kit ligand (SCF)[33] (supplemental Fig. 6a) and absence of c-Kit expression in 4T1 tumour cells (Supplemental Fig. 6b), suggested the c-Kit:SCF axis might be involved in maintaining mitochondrial metabolism in neutrophils. To investigate this, a blocking antibody which antagonised c-Kit signalling was administered to mice following establishment of the primary tumour. This treatment had no effect on tumour growth (Fig. 5a) but significantly reduced spleen weights and peripheral neutrophil numbers (Fig. 5b, c). c-Kit blockade significantly reduced neutrophil basal and maximal OCR (Fig. 5d). Accordingly, ATP synthase dependent OCR was significantly reduced together with fatty acid dependent mitochondrial function (Fig. 5e), whereas fatty acid independent mitochondrial function remained unaffected. Furthermore, blockade of c-Kit substantially reduced the ability of neutrophils

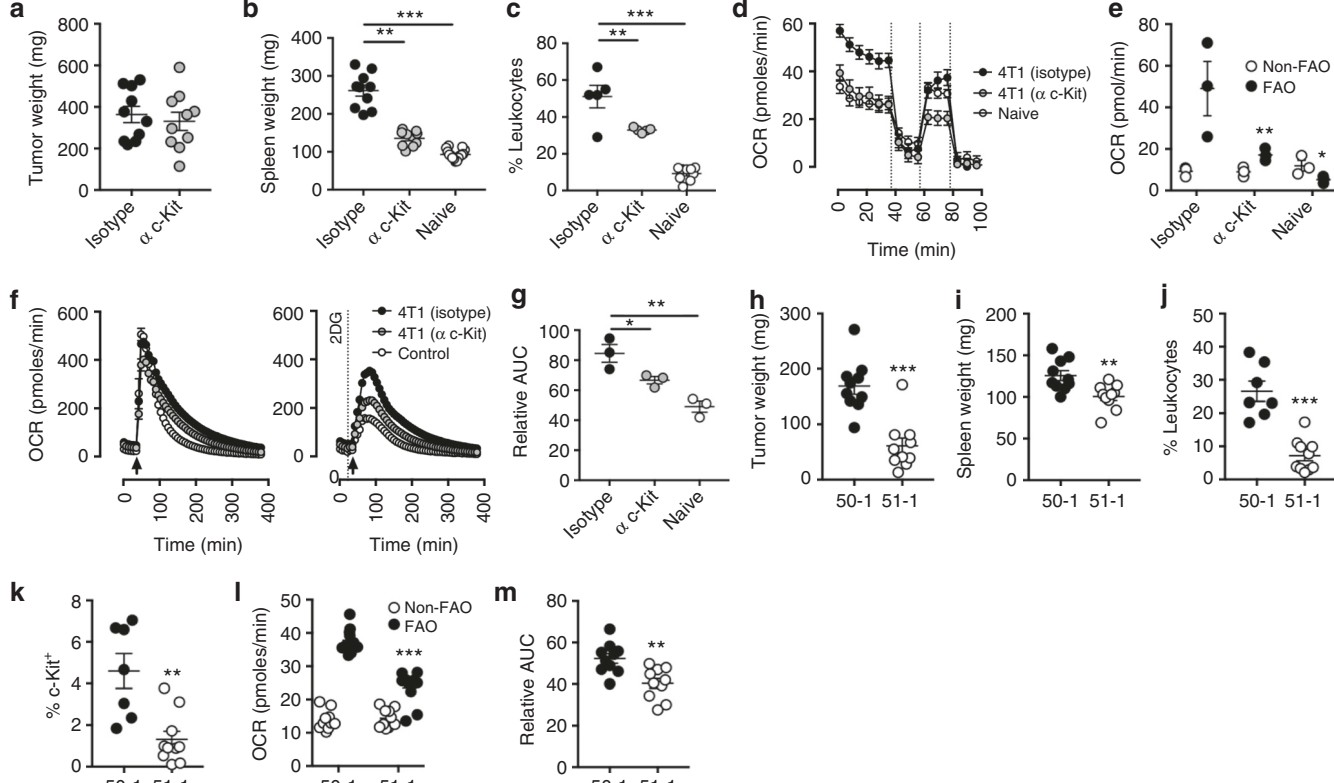

**Fig. 5** SCF-c-Kit signalling drives neutrophil metabolic adaptations. **a** Tumour weight and **b** spleen weight following administration of anti-c-Kit antibody (α c-Kit) (50 μg) or an isotype control (50 μg). Data were analysed by one-way ANOVA with Tukey's multiple comparisons displayed on graph. Tumour $n = 10$, spleen control $n = 15$, isotype and anti-c-Kit $n = 10$, from two independent experiments. **c** Percentage Ly-6G positive leukocytes in blood. Data were analysed by one-way ANOVA with Dunnett's multiple comparisons represented on graph, isotype and anti c-Kit $n = 5$, naive $n = 10$. **d** Representative neutrophil mitochondrial stress test (oligomycin (Oligo, 1.26 μM), Carbonyl cyanide-4(trifluoromethoxy)phenylhydrazone (FCCP, 660 nM), Rotenone (R, 100 nM) and antimycin A (AA, 1 μM) displaying oxygen consumption rates (OCR) with indicated treatments. **e** ATP synthase-dependent OCR displaying the proportion which is sensitive to inhibition of fatty acid oxidation (FAO) with etomoxir (100 μM). Data ($n = 3$) from two independent experiments were analysed by two-way ANOVA with Tukey's multiple comparisons indicated. **f** Representative OCR traces following stimulation with phorbol 12-myristate 13-acetate (PMA, 1 μg/ml) (arrow) in the presence or absence of 2-deoxyglucose (2DG, 100 mM) (line). **g** Area under the curve (AUC) of neutrophil OCR from data in (**f**), data shows relative OCR in the presence of 2-DG compared to PMA alone. Data ($n = 3$) from two independent experiments. Data were analysed by one-way ANOVA with Dunnett's multiple comparisons indicated. **h** Tumour weights and **i** spleen weights following injection of 4T1 with CRISPR silencing of kitl (51-1) or a non-silenced control (50-1). Data ($n = 10$) from two independent experiments, data were analysed by unpaired $t$ test. **j** Percentage of Ly-6G positive leukocytes in blood and **k** percentage of neutrophils which are c-Kit$^+$. Data ($n = 10$) from two independent experiments, data were analysed by unpaired $t$ test. **l** ATP synthase-dependent OCR displaying the proportion which is sensitive to inhibition of FAO with etomoxir (100 μM). Data ($n = 10$) from two independent experiments, data were analysed by two-way ANOVA with Sidak's multiple comparisons indicated. **m** Relative AUC of neutrophil OCR following PMA stimulation in the presence of 2-DG compared to PMA alone. Data ($n = 10$) from two independent experiments. Data were analysed by by unpaired $t$ test. $p$ Values *<0.05, **<0.01, ***<0.001. All error bars display mean ± SEM

from tumour-bearing mice to generate ROS in 2DG-simulated limited glucose (Fig. 5f, g).

To rule out effects of c-Kit blockade on responses to physiological SCF sources, and to rule out antibody mediated depletion, we generated a SCF null 4T1 cell line (supplemental Fig. 6c, d). SCF null tumours (51-1) were smaller when compared to tumours competent in SCF production (50-1) (Fig. 5h). Additionally, spleen weight, neutrophil numbers and c-Kit expression were all reduced in mice bearing SCF silenced 4T1 tumours (Fig. 5i–k). Furthermore, splenic neutrophil ATP synthase dependent OCR was reduced in mice bearing these tumours and similar to c-Kit blockade, only the proportion dependent on fatty acid oxidation was affected (Fig. 5l). Finally, this reduction in mitochondrial metabolism left neutrophils from hosts bearing SCF silenced tumour less able to maintain respiratory burst in glucose limited conditions (Fig. 5m). These data demonstrate that tumour elicited c-Kit signalling drives both the increase in neutrophil number and mitochondrial fitness resulting in an oxidative population of neutrophils which use fatty acid metabolism to maintain mitochondrial function.

**Tumour elicited neutrophils suppress T cell in low glucose.**
Neutrophils are capable of suppressing T cells via ROS production[14,36] and glucose utilisation by tumours can lower the availability of glucose in situ[22]. We therefore hypothesized that tumour elicited neutrophils might use their mitochondrial capacity to maintain suppressive activity in the glucose-depleted TME.

Injection of a luminol derivative (L-012) into the peritoneal cavity of tumour bearing mice lead to detection of MPO activity and ROS production at the tumour site whereas naive controls had no significant ROS production (Fig. 6a). Dissociation of the tumour and subsequent DCFDA staining demonstrated that ly6G$^+$ cell fractions produced significantly more ROS than the Ly6G$^-$ cell fraction and that this ROS production was a characteristic of tumour-associated neutrophils and not neutrophils isolated from other tissues such as the spleen (Fig. 6b). Interestingly, neutrophils isolated from tumours which did not produce kitl (51−1) and therefore did not display the altered metabolic phenotype (Fig. 5l), were less likely to produce ROS directly ex vivo (DCFDA$^+$) (Fig. 6c). Previous reports have suggested that neutrophils need to be in close proximity to, or even in contact with, T cells in order to deliver inhibitory ROS[37]. Indeed, neutrophils, including c-Kit$^+$ cells, were readily detected in close proximity with CD3$^+$ T cells in the TME (Fig. 6d) and furthermore tended to be in close proximity to CD4$^+$ T cells, particularly in the outer regions of the tumour, whereas CD8$^+$ T cells did not co-localise with neutrophils (Fig. 6e, f)

Co-culture assays demonstrated that activated neutrophils reduced T cell viability, proliferation and interferon-γ (IFN-γ) production (Supplemental Fig. 7a,b,c). Tumour elicited neutrophils were no more suppressive when compared to naive controls prior to stimulation. However, following stimulation, neutrophils from 4T1 bearing mice induced significantly greater T cell death (Fig. 6g). Additionally, pre-treatment of neutrophil cultures with 2DG revealed that tumour-elicited neutrophils maintained a

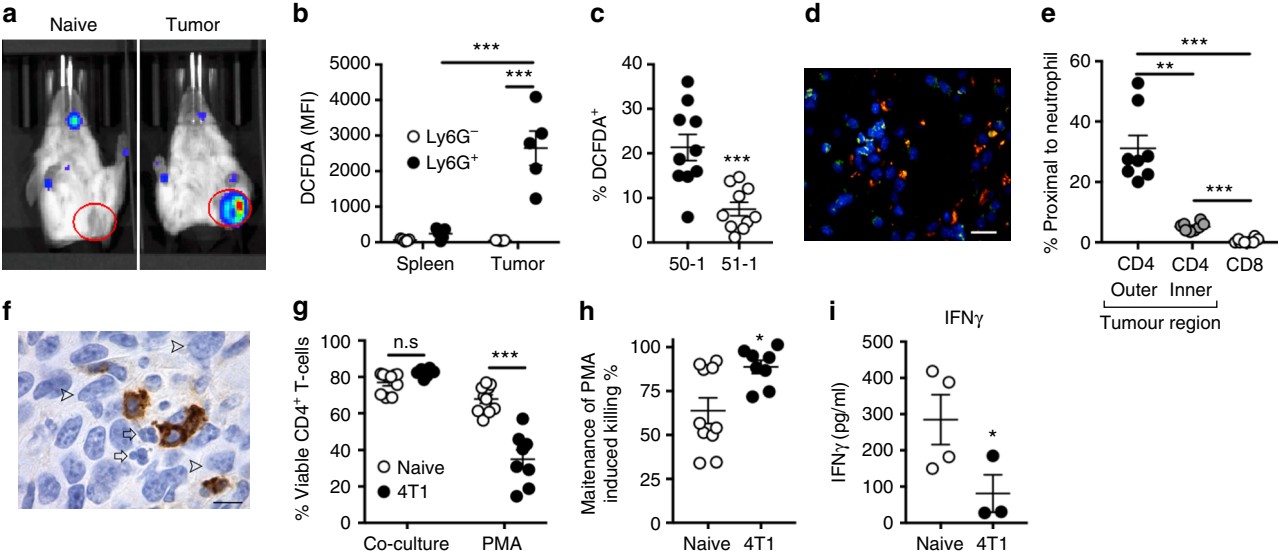

**Fig. 6** Tumour elicited neutrophils suppress T cells in low glucose. **a** Representative image at 32 min post L-012 injection showing chemiluminescence in response to myeloperoxidase (MPO) activity and the presence of H$_2$O$_2$ at the tumour site. ROIs in red show approximate location of the tumour and corresponding region on the naive mouse. **b** DCFDA fluorescence of tumour neutrophils (LY-6G$^+$) compared to Ly6G- cells from the tumour. Data were analysed by unpaired $t$-test, $n = 5$, from four independent experiments. **c** Percentage of Ly-6G$^+$ neutrophils which are DCFDA$^+$ isolated from kitl competent (50-1) and kitl null (51-1) tumours. Data were analysed by unpaired $t$-test, $n = 10$, from two independent experiments. **d** Representative immunofluorescent imaging of 4T1 tumour sections with GR-1 (Ly-6G/C)-red, c-Kit-yellow, CD3-green and 4′,6-diamidino-2-phenylindoledihydrochloride (DAPI)-blue as nuclei counter stain. Scale bar = 20 μm. **e** Quantified percentage of CD4 and CD8 T cells in close proximity (within 10 μm) to neutrophils. Data ($n = 4$) were analysed by one-way ANOVA with Tukey's multiple comparison post-tests indicated. **f** Representative immunohistochemistry diaminobenzidine staining of CD4$^+$ T cells in a 4T1 tumour cross section showing CD4$^+$ T cell and neutrophil proximity, identified by nuclear morphology (haematoxylin and eosin), arrows indicate neutrophils and arrow heads indicate tumour cells. Scale bar is 10 μm. **g** CD4$^+$ T cell viability following co-culture with neutrophils pre-stimulated with phorbol-12-myristate-13-acetate (PMA, 1 μg/ml) or control (co-culture). Data were analysed by two-way ANOVA with Sidak's multiple comparisons indicated, naive $n = 10$ 4T1 $n = 8$, from four independent experiments. **h** CD4$^+$ T cell killing following co-culture with PMA-activated splenic neutrophils stimulated in the presence of 2-deoxy glucose (2DG, 100 mM) relative to PMA alone. Data were analysed by unpaired $t$-test, naive $n = 10$ 4T1 $n = 8$, from four independent experiments. **i** Interferon-γ (IFNγ) production from co-culture of T cells and PMA activated neutrophils in the presence of 2DG from spleens of naive ($n = 3$) and 4T1-bearing mice ($n = 4$). Data from two independent experiments were analysed by unpaired $t$-test. $p$ Values *<0.05, **<0.01, ***<0.001. All panels error bars display mean±SEM

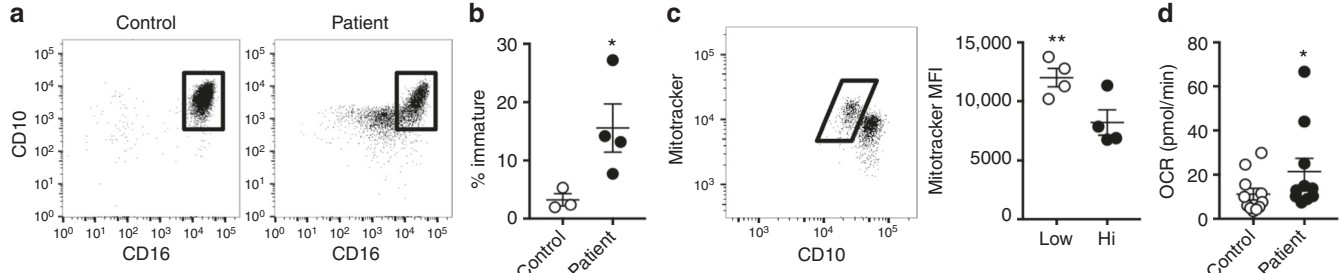

**Fig. 7** Cancer patient neutrophils possess increased OXPHOS. **a** Representative flow cytometry plots of peripheral blood neutrophils displaying CD10 and CD16 expression from healthy controls and ovarian cancer patients. The box indicates mature neutrophils. **b** Quantification of immature neutrophils from multiple patients scored as a percentage of CD14− CD15+ cells which do not express the CD10hi CD16hi marker profile associated with maturity. Data were analysed by unpaired t-test control n = 3, cancer patients n = 4. **c** Representative flow cytometry plot (left) of peripheral human neutrophils' CD10 expression and mitotracker green fluorescence. The box indicates immature CD10mid neutrophils. Quantification (right) of mitotracker median fluorescent intensity (MFI) from CD10 mid and hi peripheral neutrophils. Data (n = 4) were analysed by paired t-test. **d** ATP synthase dependent oxygen consumption rates (OCR) of peripheral blood neutrophils from healthy controls and ovarian cancer patients. Data from four independent experiments (healthy controls n = 11 and cancer patients n = 10). Data were analysed by two-way ANOVA with significance between controls and patients indicated on graph. p Values *<0.05, **<0.01. All panels error bars display mean ± SEM

greater proportion of CD4+ T cell killing, following the inhibition of glucose metabolism (Fig. 6h). Although activated neutrophils reduced the viability of CD8+ T cells, this effect was not maintained following 2DG treatment, suggesting that CD8+ T cells are less susceptible to low level neutrophil activity (Supplemental Fig. 7d). Finally, we found that IFN-γ production by T cells was significantly reduced when cultured in the presence of activated neutrophils from tumour bearing mice during 2DG treatment (Fig. 6i). Together these data strongly suggest that 4T1 elicited oxidative-neutrophils can maintain their enhanced T cell suppression even in conditions where glucose utilisation is limited, whereas glycolytic-neutrophils from naive mice cannot.

**Cancer patient neutrophils possess increased OXPHOS.** To add context to our animal model data, we assessed whether oxidative metabolism might be present in neutrophils from cancer patients. Analysis of peripheral blood neutrophils (CD14−, CD15+) revealed that, compared to healthy donors, patients harbouring ovarian cancers had an increased proportion of neutrophils with intermediate expression of CD10 and CD16; characteristics previously reported as indicative of immature phenotype[38,39] (Fig. 7a, b). Similar to our findings in mice, immature CD10int neutrophils possessed greater mitochondrial content (Fig. 7c) and concordantly, cancer patient neutrophils possessed greater ATP synthase dependent OCR compared to healthy controls (Fig. 7d). Together these data suggest, that similar to our observations in 4T1 bearing mice, in cancer patients, neutrophils with enhanced mitochondrial mass and oxidative metabolic phenotype accumulate in peripheral blood, potentially endowing them with the fuel plasticity and the ability to suppress immunity within the limiting glucose environment of tumours which we have identified in mice (Fig. 8).

## Discussion

Neutrophils are often discounted as a homogenous, short-lived and one-dimensional population, with many studies ignoring their importance in numerous physiological processes. This dogma is now being challenged, particularly in cancer[16,27]. In this study, we characterise the mitochondrial activity of neutrophils and find that when isolated from different tissue niches, neutrophils possess distinctly different metabolic phenotypes. In agreement with previous studies[23], we find that mitochondrial function is progressively lost as neutrophils mature, lose c-Kit expression and are released to the circulation and distant tissues

such as the spleen. In contrast, the mitochondria rich phenotype of cancer-associated neutrophils is maintained by c-Kit signalling, as a result of tumour derived SCF[33]. Our finding that c-Kit signalling supports mitochondrial mass, function and fatty acid oxidation is consistent with a previously reported role[40]. Mitochondria rich neutrophils are able to employ their respiratory capacity to support the generation of ROS during the stress of respiratory burst. These cells maintain the ability to produce ROS in conditions where glucose utilisation, and therefore pentose phosphate pathway (PPP) derived NADPH, is limited. The result is tumour-promoted maintenance of neutrophil populations capable of maintaining NOX-derived ROS and suppression of T cells, even in nutrient limited conditions, such as the low glucose environment of advanced tumours.

Recent studies from our laboratory and others have demonstrated that macrophages also depend on mitochondrial support for ROS generation during responses to pathogens[30,31,41]. These data showed that macrophage mitochondria favour the use of complex II, succinate dehydrogenase (SDH), whilst uncoupling their ETC from OXPHOS to directly drive substantial ROS generation by complex III. However, the data we present here suggests that neutrophils utilize their mitochondria to support ROS generation via a different mechanism, where OXPHOS does not become uncoupled from the ETC, and complex V (ATP synthase) activity is required to support glucose-independent respiratory burst. However, despite effects on ATP production in bone marrow-derived neutrophils, mitochondrial inhibition did not similarly affect tumour-elicited splenic neutrophils. Furthermore, ATP levels were unaffected by disruption of fatty acid metabolism, which is discordant with effects on respiratory burst. Together these data suggest that maintenance of ATP levels does not fully account for the mechanism by which neutrophils maintain their respiratory burst in conditions where glucose metabolism is inhibited.

Previous reports have demonstrated that mitochondria can act as a direct source of ROS in human neutrophils[32,42]. In agreement with this we find that mice deficient in the NOX subunit p47 are able to produce minimal ROS in response to PMA using DCFDA, however, we could not identify production of ROS with APF (Supplemental Fig. 3f). This is likely due to the different ROS species detected by these reagents. DCFDA detects $H_2O_2$ that can be generated either by NOX2 or the ETC. However, APF also detects other ROS and RNS species, such as hypochlorite, that are generated downstream of NOX2 activation, by the enzyme myeloperoxidase. Additionally, this PMA-induced ROS coincided

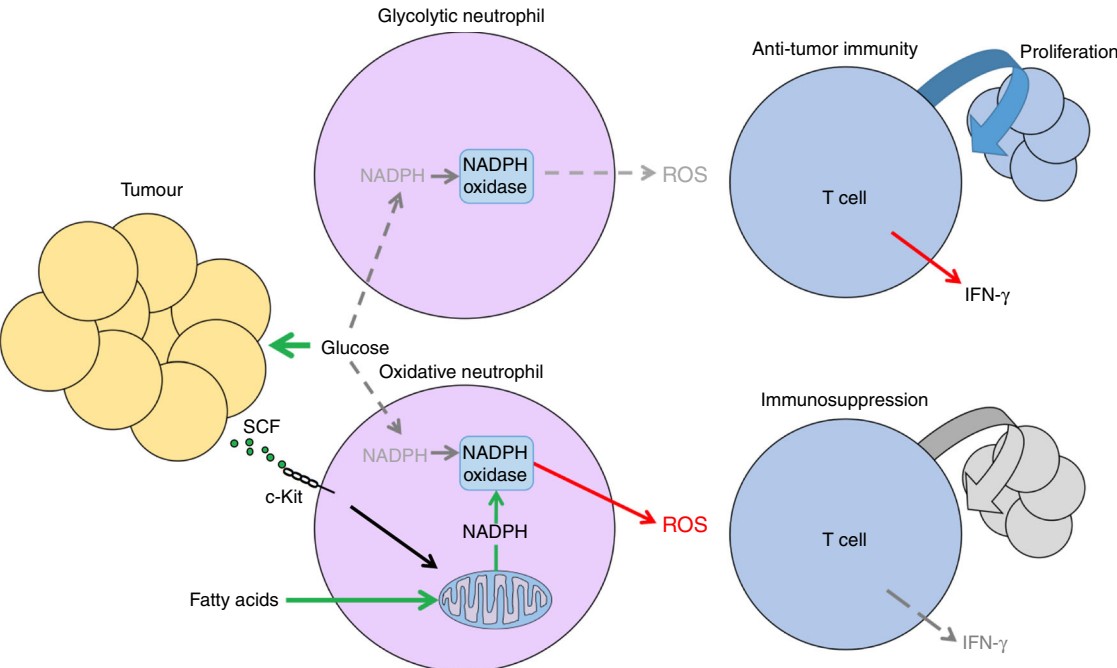

**Fig. 8** Tumours promote metabolically adapted suppressive neutrophils. Neutrophils produce ROS which is capable of disrupting CD4+ T cell viability and function in the tumour microenvironment. Glycolytic neutrophils rely on glucose to maintain production of ROS and thusly in the glucose-deprived tumour-microenvironment (TME) glycolytic neutrophil activity is predicted to be limited. However, 4T1 tumours elicit mitochondria rich oxidative-neutrophils through aberrant stem cell factor (SCF)/c-Kit signalling. Oxidative neutrophils are able to maintain NADPH-oxidase dependent ROS production in the absence of glucose utilisation through fatty acid dependent mitochondrial function to maintain NADPH levels. This c-Kit dependent mechanism allows neutrophils to overcome nutrient limitations and suppress anti-tumour immunity in the glucose deprived TME

with increased mitochondrial activity in $p47^{-/-}$ neutrophils, suggesting this organelle is the source of the minimal ROS detected. Therefore, our evidence clearly supports the indirect involvement of the ETC in ROS generation.

We show that $p47^{-/-}$ neutrophils could not engage in significant ROS production and, in the absence of glucose metabolism, NOX inhibition disrupted respiratory burst. Furthermore, NADPH levels correlated with ROS production and were significantly affected by either ETC inhibition or disruption of fatty acid metabolism. These data strongly suggest that oxidative neutrophils are able to maintain NOX activity independently of glucose by deriving NADPH via an alternative mitochondria-dependent source. Neutrophil NADPH levels have been reported to be highly dependent on glucose metabolism through the enzyme glucose-6-phosphate dehydrogenase, the first step of the PPP which is vital for neutrophil respiratory burst[43,44]. There are two possible candidates for mitochondria-dependent NADPH production. Malic enzyme 1 (ME1) which converts malate to pyruvate and isocitrate dehydrogenase (IDH1) converts isocitrate to α-ketoglutarate, both reducing NADP+ and yielding NADPH[45]. Both of these enzymes are cytosolic, but are dependent on TCA activity and subsequent export of citrate to the cytoplasm through the citrate–malate exchanger. Interestingly, macrophages have been reported to use this mechanism to maintain NADPH during glucose deprivation[46]. Regardless of its source, mitochondria-dependent NADPH allows oxidative neutrophils to support NOX activity from a variety of cellular fuels, giving oxidative neutrophils a functional advantage particularly in low glucose environments.

Despite their metabolic distinctiveness, neutrophils from 4T1-bearing hosts did not show altered expression of other surface markers and many remain c-Kit−. Furthermore, although immature nuclear morphology was increased in 4T1 neutrophil populations, it did not correlate with expression of c-Kit,

suggesting immature nuclear morphology is simply a hallmark of increased neutrophil turnover rate to maintain high circulating numbers, as seen in emergency granulopoiesis[47]. Finally, activated neutrophils from naive mice were also able to induce T cell death, albeit to a lesser extent than tumour-elicited neutrophils, suggesting that, in general, neutrophil activation is counter-productive to T cell function. We therefore suggest that, as opposed to generating an immunosuppressive population, tumours such as 4T1 may simply induce immune dysregulation, promoting neutrophil activation to inhibit adaptive anti-tumour immune responses. Furthermore, given previous reports of c-Kit+ progenitors and neutrophils with increased lifespans in the periphery during infection[48], it is possible that c-Kit+ mitochondria rich neutrophils can exist outside of the cancer setting and may represent a physiological subset vital for host defence to pathogens.

There are many different proposed mechanisms for neutrophil/MDSC-mediated immune suppression, including arginase expression[49], sequestration of extracellular cysteine[50] and IL-10 production[51]. However, ROS production has most often been reported to be the major contributor to immunosuppression by neutrophils in the tumour setting[14,36]. Additionally, MDSC have been reported to be more suppressive when derived from the tumour site[52,53] and, until now, it has been difficult to reconcile the described role of neutrophil ROS in immunosuppression with the reported dependency of these cells on glucose, a fuel widely described as being limiting in the TME[22]. Our data demonstrate that the likely explanation lies in the unexpected heterogeneity of neutrophil metabolic programming. Clearly, different neutrophil populations can have dramatically different metabolic states that permit them to adapt to niches with different fuel availability. Indeed, silencing of kitl in tumours reduced both neutrophil oxidative metabolism and their activity at the tumour site. Thus, it appears that cancers have apparently subjugated an otherwise

normal process of physiology, through release of factors such as SCF, to force the accumulation of mitochondria rich cells[26] in the glucose poor TME[22], thereby circumventing this nutrient limitation. Interestingly, human cancers also appear to promote accumulation of mitochondria rich neutrophils, suggesting that tumour elicited metabolic adaptions in neutrophils may also play a role in human disease. Targeting of this function may prove an attractive therapeutic approach for patients with high neutrophil to lymphocyte ratio and poor prognosis. Furthermore, we suggest that this heterogeneity in neutrophil metabolism is likely to play a role in multiple inflammatory disease states which recruit immature neutrophils, such as lupus[54].

## Methods

**Reagents**. All reagents were from Sigma unless otherwise stated. Rotenone, antimycin A, 2-deoxyglucose (2-DG), oligomycin, glutamate (pH 7.4 with NaOH), luminol, dodecyltrimethylammonium chloride (DTAC), atpenin (Cayman Chemicals), etomoxir, PMA, mitotracker green, CMX-ROS, Hoechst 33342, DCFDA, APF, TMRE (Thermo-Fisher).

**Mice**. C57BL/6J and NOX deficient $p47^{-/-}$ mice were maintained and bred in the Frederick National Laboratory Core Breeding Facility. NOX deficient $p47^{-/-}$ mice were a kind gift from Dr. Steven Holland (National Institute of Allergies and Infectious Diseases, Bethesda, USA). Balb cJ mice were purchased from the Jackson Laboratory. Mice were used in accordance with an approved protocol by the NCI Frederick Institutional Animal Care and Use Committee (Permit Number: 000386). Experiments on C57BL/6J were carried out in male mice, Balb cJ experiments were carried out in female mice.

**Primary cell preparation and purification**. Standard neutrophil isolation was carried out by magnetic purification. Cells were blocked with 4 µg/ml α-FcγIII (2.4G2, in house) in wash buffer (phosphate buffered saline (PBS) with 5 mM ethylenediaminetetraacetic acid (EDTA), 0.5% BSA) for 5 min on ice, before addition of 20 µg/ml α-Ly-6G-Biotin (1A8, Biolegend) for 30 min on ice. Cells were then magnetically sorted with streptavidin micro-beads and LS columns as per manufactures instructions (Miltenyi Biotec). Purities were typically >97% for bone marrow neutrophils, in spleen purities ranged from 70 to 95%. For high purity sorts required for metabolite profile or protein isolation, neutrophils were further purified by fluorescent associated cell sorting by Ly6G$^+$, purities obtained were >95%. For c-Kit$^{+/-}$ neutrophil isolations, bone marrow neutrophils were extracted as described above and then subsequently sorted by FACS for presence of c-Kit or CXCR2 receptors, purities obtained were >95%. T cells were purified from spleens and lymph nodes of naive mice and purified by Miltenyi Biotec pan T cell kit II negative selection kit as per manufacturer's instructions, purities obtained were >97%. Human neutrophils were purified from peripheral blood using Miltenyi Biotec Macsxpress Neutrophil isolation kit as per manufacturer's instructions, purities obtained were >90%. Human blood was obtained from healthy volunteers who were recruited through the National Cancer Institute-Frederick research donor program (approval number 16-003) and provided written informed consent. All users of human materials were approved and appropriately trained. Whole blood from cancer patients was collected on a study, conducted in accordance with the Declaration of Helsinki, approved by the institutional review board at the National Cancer Institute, and registered with Clinicaltrials.gov (NCT00034216). All patients provided written, informed consent at study enrolment.

**Extracellular flux analysis**. Neutrophil adherence was achieved by plating a suspension of sorted neutrophils in seahorse assay media with 2 mM glutamine and 25 mM Glucose and spinning at the lowest acceleration to 45×$g$ followed by natural deceleration. Sorted neutrophils were seeded at $0.4 \times 10^6$ cells per well and incubated for 1 h at 37 °C with no CO$_2$. XF analysis was performed at 37 °C with no CO$_2$ using the XF-96e analyser (Seahorse Bioscience) as per manufacturer's instructions. Port additions and times were used as indicated in the figures.

**Flow cytometry**. Assessment of active mitochondrial mass was performed by staining with TMRE (40 nM) or mitotracker green (5 nM) in complete sea horse media (2 mM glutamine and 25 mM glucose) for 20 min at 37 °C. Excess mitochondrial dye was removed by washing and cells were further stained for surface markers and acquired by flow cytometry. ROS was measured by incubation with either DCFDA (5 µM) or APF (5 µM) in complete sea horse media (2 mM glutamine and 25 mM glucose) for 20 min at 37 °C. Neutrophils then received stimulation with PMA (1 µg/ml) for a further 20 min, following which cells were washed to remove excess dye and further stained for surface markers and acquired by flow cytometry. Cells were transferred to flow cytometry tubes and analysed by a BD LSRII or BD Fortessa flow cytometer for fluorescent analysis. Doublets and debris were gates out before quantification of median fluorescent intensities with FlowJo

(FlowJo, LLC). See Supplementary Table 1 for information on antibodies used for flow cytometry.

**Immunoblotting and in gel activity assays**. Cellular fractionation into cytosol and intact mitochondria was performed as follows[55,56]. Briefly, cytosolic fractions were isolated after permeabilisation with a buffer containing 0.1% digitonin in 210 mM mannitol, 20 mM sucrose and 4 mM HEPES. The supernatants after the first centrifugation step at 700×$g$ for 5 min were subjected to 20,000×$g$ for 15 min The supernatants after the second centrifugation step were saved as cytosolic (soluble) fractions. The crude preparation of mitochondria isolated by differential centrifugation was lysed in lysis buffer I containing 50 mM BisTris, 50 mM NaCl, 10% w/v Glycerol, 0.001% Ponceau S, 1% Lauryl maltoside, pH 7.2, protease and phosphatase inhibitors.

The NativePAGE Novex Bis-Tris gel system (Thermo Fisher Scientific) was used for the analysis of the mitochondrial respiratory chain complexes, with the following modifications: only the Light Blue Cathode Buffer was used; 20 µg of mitochondrial protein extracts were loaded/well; the electrophoresis was performed at 150 V for 1 h and 250 V for 2 h. For the native immunoblots, PVDF was used as the blotting membrane. The transfer was performed at 25 V for 4 h at 4 °C. After transfer, the membrane was washed with 8% acetic acid for 20 min to fix the proteins, and then rinsed with water before air-drying. The dried membrane was washed 5–6 times with methanol (to remove residual Coomassie Blue G-250), rinsed with water and then blocked for 2 h at room temperature in 5% milk, before incubating with the desired antibodies diluted in 2.5% milk overnight at 4 °C. In order to avoid strip and reprobing of the same membrane, which might allow detection of signals from the previous IBs, samples were loaded and run in replicates on adjacent wells of the same gel, and probed independently with different antibodies.

In-gel Complex I, Complex II and Complex IV activities were performed as follows[56–58]. For Complex I activity, after resolution of the respiratory chain complexes by BN-PAGE, the gel was incubated with 0.1 M TrisCl, pH 7.4, containing 1 mg/ml nitrobluetetrazolium chloride (NBT) and 0.14 mM NADH at room temperature for 30–60 min For complex II, detection of succinate CoQ-reductase activity (SQR) (CoQ-mediated NBT reduction) was performed by incubating the gel for 30 min with 84 mM succinate, 2 mg/ml NBT, 4.5 mM EDTA, 10 mM KCN, 1 mM sodium azide and 10 µM ubiquinone in 50 mM PBS, pH 7.4. For complex IV, the gel was incubated in 50 mM phosphate buffer pH 7.4 containing 1 mg/ml DAB (3,3′-diaminobenzidine) and 1 mg/ml cytochrome c at room temperature for 30–45 min

**Luminescence assays**. Magnetically sorted neutrophils were seeded at 0.25–0.3 × 10$^6$ cells per well of a 96-well luminescence plate in complete seahorse media (glucose 25 mM and L-glutamine 2 mM) and incubated for 30 min at 37 °C. For H$_2$O$_2$ measurements 400 µM luminol was added to the media and cells were stimulated for 20 min with 2DG (100 mM), Rotenone (100 nM) and antimycin A (1 µM) or etomoxir (100 µM) following which cells were stimulated with PMA (1 µg/ml) and immediately measured for luminescence in a 96-well luminescence plate reader and luminescence readings were measured every 6 min for 6 h. For ATP and NADPH measurements, cells were lysed in passive lysis buffer (Promega) or 0.1 M NaOH with 0.5% DTAC for ATP or NADPH determination respectively. Relative ATP was determined using the ATP assay kit (Abcam/Thermo-Fisher) and NADPH was determined using the NADPH-Glo assay kit (Promega) as per manufacturer's instructions.

**4T1 tumour model**. The 4T1 cell line was maintained from frozen stocks at low passage number and cultured using complete DMEM media (10% foetal calf serum, 2 mM glutamine and penicillin/streptomycin 100 U/ml/100 µg/ml). Female Balb/cJ mice received a single sub cutaneous injection of $1 \times 10^5$ 4T1 cells or sterile PBS into the 3rd mammary pad. Tumour size was determined using calipers. The maximum tumour size permitted by the NCI-Frederick IACUC, and adhered to in this study, was 20 mm at the largest diameter. Mice were monitored daily and euthanised if they displayed any signs of toxicity (rough hair coat, laboured breathing, lethargy, rapid weight loss, difficulty in obtaining food or water) or if tumours ulcerated, became necrotic or infected. Mice were euthanized 14 days post injection and tissues such as spleen and tumour were removed for experimentation. C-Kit blocking experiments were performed by intraperitoneal injections of anti-c-Kit or appropriate isotype control (50 µg) on daily for commencing from day 10.

**CRISPR-cas9 silencing**. CRISPR guide RNAs (sgRNA) were designed to target the SCF gene kitl located on chromosome 10 of the mouse genome using the sgRNA scorer 2.0 and candidates were selected based on their high predicted activity and lack of off-target effects. Selected sgRNA were cloned into pX458 cloning vector containing a Cas9-2A-GFP by ligating two annealed oligos following digestion with BbsI restriction enzyme. 4T1 cell lines were transfected by CRISPR plasmid using x-tremeGENE (Sigma-Aldrich) transfection reagent at a ratio of 3:1 (90 µl X-tremeGENE : 30 µg of plasmid) in 1000 µl of serum free Optimem and incubated for 15 min at room temperature. Transfection suspension was added to 10 ml of RPMI and added directly to cells. Following transfection, GFP$^+$ 4T1 cells were found to be typically 4% of total cells and were purified by fluorescent associated

cell sorting and placed in single cell cultures. Cultures were first screened for SCF production using ELISA (Sigma-Aldrich), selected candidates had editing efficiencies confirmed by Illumina Miseq sequencing for non-homogenous end joining (NHEJ) induced mutation rates. CRISPR silenced 4T1 cell lines were compared to 4T1 transfected cell lines which failed to silence kitl gene.

**L-012 luminescence**. 2D chemiluminescence imaging was performed on tumour bearing mice using the IVIS SPECTRUM scanner (PerkinElmer Inc., Waltham, MA) in the supine position. 250 µl of L-012 (Wako Chemicals) in PBS was administered to each mouse IP (60 mg/kg). Mice body temperature was maintained at 37 °C during the procedure with a heated pad located under the anaesthesia induction chamber, imaging table, and post procedure recovery cage. Anaesthesia was initially set at 3% isoflurane with filtered (0.2 µm) air at 1 l/min flow rate for 3–4 min and then modified for imaging to 2% with $O_2$ as a carrier with a flow rate 1 l/min. Dynamic 2D images were acquired at every 2 min for the total of 50 min with the following parameters: excitation filter-blocked, emission filter-open, f/stop 1, medium binning (8×8) and 2-min exposure. Acquisition and analysis was performed via vendor specific software Living Image (version 4.3.1) (PerkinElmer Inc, Waltham, MA).

**Confocal microscopy**. For neutrophil mitochondrial images, isolated neutrophils were incubated in complete sea horse media (2 mM glutamine and 25 mM glucose) for 20 min at 37 °C with mitotracker CMX ROS (100 nM). Cells were washed in PBS and incubated with Hoechst (2 µg/ml) for 20 min with 5 °C after which cells were washed in PBS. Neutrophils were placed in a cytospin cartridge and spun at 800 rpm and spun for 3 min. Cells were then fixed in 4% PFA for 20 min, washed and a coverslip was added. A Zeiss UV-510 confocal microscope and 63× oil immersion objective lens was used to capture images using differential interference contrast, 405 and 561 laser paths (pinhole size = 1 airy unit).

Tumour sections were processed fresh-frozen sections and were washed with PBS, mild-fixed with acetone at 4 C, permeablised with 0.05% Triton in PBS and blocked with 5% BSA in PBS. Sections incubated at room temperature for 1 h with the following antibodies (1:100, all from Biolegend): FITC anti-mouse CD3, PerCP/Cy5.5 anti-mouse GR-1, APC anti-mouse c-Kit and counterstained with DAPI. Images were analysed using Fiji (ImageJ).

**Immunohistochemistry analysis of 4T1 tumours**. Tumours were excised, washed and cut in half. One half was fixed by submersion in 10% neutral buffered formalin for 24 h and paraffin-embedded. The other half was submerged in OCT compound, mounted and snap frozen in a dry ice/methanol slurry. For immunohistochemistry analysis, 10 µm paraffin sections were deparaffinized, blocked and stained with either a biotinylated rat anti-mouse CD4 monoclonal antibody (1:250, eBioscience) or CD8a monoclonal antibody (1:250, eBioscience). EDTA or Citrate was used as antigen retrieval agent. Images were taken using an Olympus BX40 microscope.

**Primary metabolite analysis**. Fluorescent associated cell sorted peritoneal neutrophils ($5.0–10 \times 10^6$) were pelleted and snap-frozen in liquid nitrogen. Samples were further processed and analysed at the West Coast Metabolomics Center (University of California, Davis). Briefly, samples were re-suspended with 1 ml of extraction buffer (37.5% degassed acetonitrile, 37.5% isopropanol and 20% water) at −20 °C, centrifuged and decanted to complete dryness. Membrane lipids and triglycerides were removed with a wash in 50% acetonitrile in water. The extract was aliquoted into two equal portions and the supernatant dried again. Internal standards C08-C30 fatty acid methyl esters were added and the sample derivatized by methoxyamine hydrochloride in pyridine and subsequently by N-methyl-N-trimethylsilyltrifluoroacetamide for trimethylsilylation of acidic protons. GC-TOF analysis was performed by the Agilent GC6890/LECO Pegasus III mass spectrometer. Samples were additionally normalized using the sum of peak heights for all identified metabolites.

**T cell suppression assay**. Purified neutrophils were plated in 96-well U-bottomed plates at $0.125 \times 10^6$ cells per well in RPMI supplemented with (Pen/Strep, 10% FCS, 2 mM L-glutamine, 1 M HEPES, 0.1 mM nonessential amino acids) and adhered by spinning at the lowest acceleration to 45x*g* followed by natural deceleration. Neutrophils were incubated for 30 min–1 h at 37 °C followed by a 20 min stimulation with 2DG (100 mM) or vehicle control, after which cells were stimulated with PMA (1 µg/ml) for 5 min. Neutrophil media was removed and cells washed with fresh RPMI. Purified T cells were overlayed at $0.25 \times 10^6$ and co-cultured for either 18 h for viability and IFNγ measurements (Fig. 6) or 72 h for proliferation index (supplemental Fig. 7). T cell viability was assessed by Sytox Green (ThermoFisher) uptake and subsequent flow cytometry. BD cytometric bead array kit was used to quantify Interferon-γ in supernatant taken 16 h after T cell suppression assay, and was used as per manufacturer's instructions. Proliferation index was calculated by pre-culturing T cells with CellTrace Violet Cell Proliferation Kit (ThermoFisher) by gating cells by their number of division and assigning a generation number, where no division is 0 and 1 division is 1 and so on. The following equation was used to calculate the proliferation index (PI), the average number of divisions a dividing cell will go through, where $i$ is the

generation number and $n$ is the number of events in that generation number[59,60].

$$PI = \frac{\sum_1^i i \times \frac{N_i}{2^i}}{\sum_1^i \frac{N_i}{2^i}} \quad (1)$$

**Statistical analysis**. Statistical analysis was performed by GraphPad Prism 6, with the tests used indicated in the figure legend. Data log transformed where appropriate before analysis. $*p < 0.05$, $**p < 0.01$, $***p < 0.001$. All error bars represent the mean ± the standard error of the mean (SEM).

## Data availability

All data is freely available from authors upon request.

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

## Acknowledgements

Thank you to Laura Coffin for assistance in luminol measurement of $H_2O_2$ and Megan Karwan for assistance in technical work involving mouse models. This work has been funded with federal funds from the National Cancer Institute, National Institutes of Health, Intramural Research Program, USA. L.C.D. is funded by the Henry Wellcome Trust, UK (WT103973MA).

## Author contributions

C.M.R. conceived and designed the project conducted and analysed experiments, interpreted the data and wrote the manuscript. L.C.D. conducted experiments, assisted with interpretation of data and critically appraised the manuscript. J.J.S., N.M., N.L.P. and E.M.P. conducted experiments, analysed and assisted with interpretation of data. M. G.C. and C.A. analysed and assisted with interpretation of data. J.L. and C.M.A. sourced and collected cancer patient peripheral blood for analysis. J.M.W., T.A.R. and S.K.D. helped critically appraise and edit the manuscript. D.W.M. facilitated the development and progression of the project, assisted with interpretation of the data and wrote the manuscript.

## Additional information

**Competing interests:** The authors declare no competing interests.

