## [Peer Review File · Nature Communications]

Reviewers' comments:

Reviewer #1 (Remarks to the Author):

In the manuscript "Tumour-elicited neutrophils engage mitochondrial metabolism to circumvent nutrient limitations and maintain immune inhibition", Rice et al. show that a subset of neutrophils (defined originally as Ly6G+cKit+, but more generally as lineage "immature") exhibit significant mitochondrial metabolic activity which supports their classic effector functions such as respiratory burst. The authors show that this residual mitochondrial activity is sufficient to help drive neutrophil function in glucose-restricted conditions in vitro, and correlate these findings in both a mouse tumor model and human cancer patients, contexts where this finding is likely to be highly relevant. The in vitro work frequently uses a complex combination of metabolic inhibitors that the authors use adeptly to arrive at some important and nuanced discoveries. The idea that mitochondrial metabolism can support NADPH levels for NADPH oxidase activity in glucose-restricted conditions is particularly delightful. The data showing that tumor-bearing mice preferentially elicit this subset of neutrophils is highly interesting and important for many fields of study. The authors' key findings (of the importance of mitochondrial metabolism for neutrophil function) help challenge the dogma in the neutrophil community regarding the belief that neutrophils are almost exclusively glycolytic and expand our understanding of this cell type. In general, the manuscript is well-written, the experiments are well-controlled, and the authors' conclusions are supported by the data. The authors show appropriate restraint in discussing the impact of their correlative human data, yet this remains a nice clinically-relevant finale to the work. This novel work is likely to be of significant interest to the tumor immunology, neutrophil immunology, and the general immunometabolism communities. The statistical analyses employed are appropriate and the level of detail in the methodology section is sufficient to allow researchers to reproduce the studies accordingly. Overall this is a strong study of significant importance to the field. A few concerns/suggestions to improve the manuscript are enumerated below:

- 1) In general, it is not clear how definitive the authors' claim regarding the neutrophil identity of their cells of study are (for example, there is transient Ly6G expression of monocyte precursors in the bone marrow). This concern does not in any way diminish the importance of the findings. However, I would recommend more explicitly defining cell population in figures. For example, the y-axis for Figure 1D would be more precise as "% Ly6G+cKit+ cells" as opposed to "% Neutrophils".
- 2) It is not entirely clear from my reading, of the manuscript if the Rot/AA treatment group in Figures 2C and 2E contradict each other. This should be resolved or better explained for reader clarity.
- 3) For the cKit blockade studies in Figure 4, the authors claim "we blocked c-Kit signalling", suggesting that antibody treatment was merely blocking the signaling of the target neutrophil population. It seems pretty clear from the data presented that the target population is largely being depleted from circulation, which explains the data since the remaining neutrophils are the conventional "mature" neutrophils lacking in significant mitochondrial capacity. Again, this does not diminish the findings in my view, but I would recommend that the authors finesse the language a bit to more accurately describe that this is essentially a subset depletion experiment and not a signal blockade experiment. If the authors believe that their original interpretation is correct, more explicit demonstration to that effect would be useful for the readers.

Reviewer #2 (Remarks to the Author):

Rice and colleagues presented the results of the study where they show that immature neutrophils have the capacity for oxidative mitochondrial metabolism. In limited glucose environment, neutrophils use mitochondrial fatty acid oxidation to support NADPH---oxidase dependent ROS production. In 4T1 tumor bearing mice, mitochondrial fitness is enhanced in splenic neutrophils and is driven by c-Kit signaling. Tumor-elicited neutrophils are able to maintain ROS production and suppression of T-cells when glucose utilization is restricted. Cancer patients peripheral blood neutrophils also display increased immaturity, mitochondrial content and subsequent oxidative phosphorylation.

Overall concept of the study is interesting. Although understanding of OXPHOS involved in MDSC activity has started to emerge presented data could potentially advance the field. Authors conclusions are exciting. The problem, many of them are not supported by the data.

On less important point. Most of the experiments in Figure 1 are convincing and don't raise issues. However, the main statement regarding relatively mature vs. immature state of c-kit+/- neutrophils requires stronger justification. Authors made their conclusion based only on the shape of nuclei in the cells. There are number of phenotypic and biochemical characteristics that helps to identify the state of maturation of these cells. It would be important to better substantiate this conclusion.

The results in Figure 2 raised more serious concerns. Figure 2C does not show claimed differences at late time points after stimulation. More importantly, it is difficult to follow the logic of subsequent experiments. Authors basically blocked different mitochondrial enzymes and demonstrated that it resulted in inhibition of OCR in glucose independent fashion. However, this would be the result regardless whether they block glucose or not since OCR depends on mitochondrial function. Authors correctly stated at the beginning of this section that OCR does not fully reflect respiratory burst. However, they did not look at H₂O₂ production by these cells or measure crude ROS by DCFDA as they did in other experiments. In experiment with p47 KO cells authors reported some level of cytoplasmic ROS (measured by DCFDA) suggesting that mitochondria contribute superoxide. This is entirely expected. Then authors indicated that deprivation of glucose would enhance this effect. This would be interesting observation. However, surprisingly, they did not measure ROS in these experiments but just OCR. In addition, in all experiments authors did not look at any functional consequences of respiratory burst to link their findings with biological effect. As presented, Figure 2 provide very little information to support authors conclusion.

The biological significance of the observations remains unclear. In Figures 4-6 authors repeatedly stated their conclusion about the role of glucose deprivation as important functional distinction between neutrophils in naïve and tumor-bearing mice. However, they provide no evidence that such disparity in glucose level exist in bone marrow and spleen between naïve and tumor-bearing mice. It is easy to envision that their concept could work in tumor site where there is evidence of competition for glucose. However, they did not perform any direct experiments to assess function of neutrophils in tumors. There is evidence that tumor associated MDSC are more suppressive per cell basis than spleen MDSC from the same mice. Some references include: Maenhout, S.K., et al. (2014) *Int J Cancer* 134, 1077-1090. Haverkamp, J.M., et al. (2011) *Eur J Immunol* 41, 749-759. Cimen Bozkus, C., et al. (2015) *J Immunol* 195, 5237-5250. However, how this would fit to proposed concept is not clear.

Putting glucose deprivation issue aside, authors concept in Figure 8 suggest that c-kit drives more ROS production by neutrophils in tumor-bearing hosts that made them suppressive. However, authors did not present data supporting this concept.

c-kit inhibition is interesting. However, it is difficult to interpret these data the way suggested by the authors. C-kit is critically important for myelopoiesis, especially granulocytes. Inhibition of c-kit in tumor-bearing mice will dramatically reduce neutrophil population. Authors observed it in their study. It is difficult directly connect inhibition of c-kit with its effect on OCR. This may reflect selection "survival of the fittest" phenomenon.

Reviewer 1

We thank the reviewer for his favorable characterization of our work.

Comment 1.

“In general, it is not clear how definitive the authors’ claim regarding the neutrophil identity of their cells of study are (for example, there is transient Ly6G expression of monocyte precursors in the bone marrow). This concern does not in any way diminish the importance of the findings. However, I would recommend more explicitly defining cell population in figures. For example, the y-axis for Figure 1D would be more precise as ‘% Ly6G+cKit+ cells’ as opposed to ‘% Neutrophils’.”

Both Reviewer 1 and 2 had concerns regarding the identity and maturity of the bone marrow neutrophil subsets identified in figure 1 and suggested refinement of the maturation stage and relation to monocyte precursors of these neutrophils. Indeed, both neutrophils and monocytes share a common progenitor termed the granulocyte-macrophage progenitor (GMP). GMP cells also express c-Kit and although not reported to express either Ly6C or Ly6G (Delano et al., 2011) there is a possibility that transient expression of either of these receptors could lead to mistaken identity as a neutrophil or monocyte. The high density of Ly6G that we observe in our population of c-Kit^{+/−} neutrophils strongly suggests that these cells are unlikely to be monocyte precursors transiently expressing Ly6G, albeit at a lower density. Regardless, we have now included our gating strategy for Ly6G expression (new supplemental figure 1a). Additionally, in order to further investigate any potential overlap with the monocytic cell lineage we examined expression of the monocyte makers Ly6C and CD115 (MCSF-R) in our sub-populations. We find that classically identified monocytes do not overlap with our Ly6G⁺ subsets for expression of either Ly6C or MCSF-R, CD115 (new supplemental figure 1b). Moreover, as suggested, we have changed the Fig. 1d y-axis and now label these as % Ly6G⁺ c-Kit⁺ and have added discussion of this issue on page 6 of the revised manuscript (line 14).

We too initially shared the concerns of the reviewers when identifying our populations of interest. Early studies used Ly6G and c-Kit alone to identify different bone marrow-derived neutrophil populations. Using this strategy, we also identified a population with intermediate levels of Ly6G (Ly6G^{int}) that were also c-Kit⁺, which are not to be confused with the Ly6G^{hi} population we have studied. The Ly6G^{int}/c-Kit⁺ population we initially identified has an even

greater spare and maximal respiratory capacity when compared to our Ly6G^{hi} c-Kit⁺ cells (reviewer's Fig1 a-c). However, assessment of the nuclear morphology demonstrated a large proportion of cells which did not display banded or segmented nuclei and instead appear to look more similar to a myelocyte stage in neutrophil development (Pillay et al., 2013) (reviewer's Fig 1d). Additionally, flow cytometry analysis suggested significantly higher Ly6C and CD115 expression in this Ly6G^{int} population, perhaps suggesting these cells represent a population with traits closer to both subtypes and maybe a more GMP phenotype (reviewer's Fig 1e, f). These reasons, combined with the fact that we do not detect Ly6G^{int} in the blood or spleen of mice with 4T1 tumors, led us to subsequently exclude this Ly6G^{int} population from further analysis as we were concerned that they may indeed represent a monocyte/neutrophil shared progenitor population. In our study we only examine the Ly6G^{hi} neutrophils.

Comment 2.

"It is not entirely clear from my reading, of the manuscript if the Rot/AA treatment group in Figures 2C and 2E contradict each other. This should be resolved or better explained for reader clarity."

Indeed, Figures 2C and 2E represent different treatments. In Figure 2C neutrophils received a pretreatment of Rotenone/antimycin A and stimulation of PMA, which led to no difference in peak ROS production. This is different from Figure 2E, where neutrophils received pretreatment of both 2DG and Rotenone/antimycin A, which subsequently led to a complete reduction in ROS production. We apologize for any confusion and have edited the figure key to further clarify this difference.

Comment 3.

"For the cKit blockade studies in Figure 4, the authors claim "we blocked c-Kit signalling", suggesting that antibody treatment was merely blocking the signaling of the target neutrophil population. It seems pretty clear from the data presented that the target population is largely being depleted from circulation, which explains the data since the remaining neutrophils are the conventional "mature" neutrophils lacking in significant mitochondrial capacity. Again, this does not diminish the findings in my view, but I would recommend that the authors finesse the language a bit to more accurately describe that this is essentially a subset depletion experiment and not a signal blockade experiment. If the authors believe that their original interpretation is correct, more explicit demonstration to that effect would be useful for the readers."

We believed that tumor initiated SCF/C-Kit signaling was likely to be the driver of altered neutrophil metabolism in our model because, 1) anti c-Kit antibodies reversed overall metabolic alterations despite the majority of tumor-associated neutrophils remaining c-Kit negative, and 2) c-Kit has already been demonstrated to be rapidly shed from the surface of neutrophils in 4T1 bearing mice (Kuonen et al., 2012). We felt that these facts would make unlikely that we would efficiently deplete c-Kit⁺ neutrophils only. However, we agree with the reviewer that depletion is a possible cause of this metabolic reversion. Therefore, to further assess the crosstalk between tumor-derived SCF and c-Kit in neutrophils we have used CRISPR-Cas9 technology to create a 4T1 tumor which no longer expresses the ligand for c-Kit

(*kitl*)(51-1). We compared these tumors to 4T1 tumors derived from cells which received the CRISPR-Cas9 silencing plasmid but failed to delete the *kitl* gene (50-1). These new data demonstrate that *kitl* expression by the tumor is required for effective *in vivo* growth (new Figure 5h), splenomegaly (new Fig 5i) and increased neutrophil numbers (new fig 5j) and c-Kit expression (new fig 5k). Most importantly, silencing of *kitl* expression in the 4T1 tumor reduced neutrophil ATP synthase dependent OCR, specifically via fatty acid oxidation (new fig 5m) and the ability to maintain respiratory burst in conditions where glucose utilization is limited (new fig 5n). These data confirm our conclusions based on the anti-c-Kit antibody studies. Furthermore, in response to a comment from Reviewer 2, we have now included measurements of tumor neutrophil ROS and have demonstrated that neutrophils derived from tumors which no longer produce Kitl display reduced ability to produce ROS *in situ* (new fig. 6c). We thank the reviewers for pointing out this important issue in our data and prompting the experiments that strengthen our conclusions. We now discuss these changes on page 16, 17 and page 26 of the revised manuscript. Taken together, we believe these data conclusively highlight the importance of tumor initiated Kitl/c-Kit crosstalk in maintaining neutrophil mitochondrial metabolism and effector function.

Reviewer 2

We appreciate that the reviewer found our conclusions to be, “exciting” although we disagree with the reviewers contention that some may not be sufficiently supported. To strengthen the support for our conclusions we have added new data, explanations and clarifications as noted below.

The reviewer noted that, *“most of the experiments in Figure 1 are convincing and don’t raise issues. However, the main statement regarding relatively mature vs. immature state of c-kit⁺/- neutrophils requires stronger justification. Authors made their conclusion based only on the shape of nuclei in the cells. There are number of phenotypic and biochemical characteristics that helps to identify the state of maturation of these cells. It would be important to better substantiate this conclusion.”*

The reviewer raises a valid concern, also raised by reviewer 1, regarding characterization of the neutrophil populations and maturity. The use of nuclear morphology is still considered to be the gold standard of identification of developmental state neutrophil (Coffelt et al., 2016; Pillay et al., 2013; Stejskal et al., 2010) surface markers do change during development. Although we suggest CXCR2 and c-Kit are good choices, and the expression of c-Kit has already been established to denote immature neutrophils (Coffelt et al., 2015; Deniset et al., 2017; Kuonen et al., 2012), in response to the reviewer’s concern we have now further characterized our subtypes using additional maturity associated markers, CD62L, CD11b, and an immaturity marker CXCR4 (Coffelt et al., 2016) (Weisel et al., 2009). In addition to our high expression of Ly6G and lack of monocyte markers (see response to Reviewer 1, comment 1, and new supplemental figure 1a,b), we find a similar expression of CD11b and CD62L and a lack of high expression of CXCR4 (new supplemental figure 1c) when compared to Ly6G negative cells. We now discuss these data on page 6 of the revised manuscript (line 14). As nuclear morphology has previously been used to identify neutrophils at varying stages of development, we feel that our nuclear morphology data together with our surface marker expression strongly suggest the c-Kit⁺ cells assessed here are indeed band cell neutrophils, at the later stage of development before the fully mature neutrophil. Additionally, as a positive control of sorts, we have included in the reply figures the characterization of a more immature, possibly shared granulocyte/monocyte precursor (GMP) with its associated surface markers and metabolic profile, which we initially identified and rejected based on the possibility of a GMP phenotype and lack of its detection in the 4T1 bearing mouse.

The reviewer noted, *“The results in Figure 2 raised more serious concerns. Figure 2C does not show claimed differences at late time points after stimulation.”*

We regret that our initial Figure 2 did not adequately illustrate the differences at late time points after stimulation. The reduction in late H₂O₂ production, as measured by luminol, is shown in panel 2C. For our study, we chose to show the time course data, however we appreciate that with overlapping traces the data can be difficult to follow. To better display these data, area under the curve analysis has now been included in the supplementals (new supplemental 2a)

demonstrating the unaffected early peak burst and the significantly reduced late burst during Rotenone and antimycin A treatment. We have also included this for all OCR traces in figure 2. (new supplemental Fig 2b-e). We hope this better clarifies our written interpretation of the data and shows that there may be a time delay between glucose-fueled ROS and mitochondrial fueled ROS, which we discuss on page 9 of the manuscript.

The reviewer noted, *“More importantly, it is difficult to follow the logic of subsequent experiments. Authors basically blocked different mitochondrial enzymes and demonstrated that it resulted in inhibition of OCR in glucose independent fashion. However, this would be the result regardless whether they block glucose or not since OCR depends on mitochondrial function.”*

We agree that recent publications use OCR to show mitochondrial function, however this can be misleading and there remains some confusion with users of metabolic flux analysis. It must be emphasized that OCR can only be deemed mitochondrial if it is sensitive to antimycin A/ rotenone.

(https://www.agilent.com/en/products/cell-analysis/mitochondrial-respiration-xf-cell-mito-stress-test?sh_overview).

Here we show that after PMA stimulation of neutrophils the seahorse is measuring NOX activity, not mitochondrial function. Indeed, this is how respiratory burst was originally defined as the production of ROS consumes large amounts of molecular oxygen. The term is now often used to describe ROS production. The reviewer is correct that manipulation of mitochondrial function could affect this readout, however we demonstrate that the high amplitude OCR detected following PMA is an indication of oxygen consumed by NOX to produce ROS and not mitochondrial function. Firstly, the amplitude of OCR in response to PMA is unaffected following complete inhibition of mitochondrial function by rotenone and antimycin A (Panel 2a and revised text on page 8). Furthermore, NOX deficient neutrophils are unable to produce significant burst as measured by OCR (Fig. 2i), this despite enhanced mitochondrial function shown by small increases in rot/aa sensitive OCR (supplemental Fig 2g). Finally, NOX inhibition significantly blunted PMA-induced OCR increases regardless of glucose metabolism status (Figure 2j), strongly suggesting that oxygen consumption is not directly via the mitochondria, but instead is by oxygen consuming NOX complexes. Furthermore, to alleviate any concern we have supported our OCR data with direct measurement of H₂O₂ (via luminol) and detection of ROS by fluorescent probes (Fig2c, e, supplemental Fig 2f, Fig 4h). Therefore, based on all these data we contend that it is unlikely that in our subsequent panels the ablation of OCR is due purely to alterations in mitochondrial OCR, but instead due to limitations in the metabolic programming required to maintain NOX-derived burst. This is discussed on pages 22 and 23 of the manuscript. We apologize if there was confusion and we have reviewed the text and made modifications to clarify this important point.

The reviewer noted, *“Authors correctly stated... that OCR does not fully reflect respiratory burst. However, they did not look at H2O2 production by these cells or measure crude ROS by DCFDA as they did in other experiments.”*

The reviewer is correct in that OCR alone is not a direct readout for ROS production, however as discussed above we have shown that the vast majority of oxygen consumption (OCR) after PMA treatment is not linked to mitochondrial function but rather is directly dependent on NOX activity. To further support our OCR data we have used luminol measurement of H₂O₂ (Figures 2C and E) to demonstrate that ROS production directly correlates with OCR measurements (Fig 2a and d). These data are also supported by DCFDA and APF detection of ROS in supplemental figure 2f, and later in Fig 4h.

The reviewer noted, *“In experiment with p47 KO cells authors reported some level of cytoplasmic ROS (measured by DCFDA) suggesting that mitochondria contribute superoxide. This is entirely expected. Then authors indicated that deprivation of glucose would enhance this effect. This would be an interesting observation. However, surprisingly, they did not measure ROS in these experiments but just OCR.”*

Despite detection of minimal ROS by DCFDA, unlike wild type neutrophils, p47^{-/-} neutrophils did not produce ROS detectable by APF. The disparity between DCFDA and APF suggests that this could indeed be mitochondrial-derived superoxide, as opposed to other ROS species generated downstream of NOX activity. As discussed above, when NOX is present, the vast majority of OCR induced by PMA is NOX-derived as pretreatment with Rotenone and antimycin A alone does not reduce the peak of OCR induced by PMA. Therefore, the OCR measurements in neutrophils lacking NOX cannot, and were not, intended to reflect ROS production. We described the OCR increase as an increase in mitochondrial function (basal OCR) as it was sensitive to Rotenone and antimycin A. Some of this may result in H₂O₂ production, though this was not a specific question of our study. This ROS has been shown in human neutrophils to be sufficient to kill pathogens in the absence of NOX (Fernandez-Boyanapalli et al. 2014). However our data show that the mitochondria in our mouse neutrophils are not significantly contributing to respiratory burst in the absence of NOX, and that NOX is the primary site of ROS production. Further, we do not suggest that glucose deprivation enhances mitochondrial ROS, but instead we demonstrate that when glucose utilization is restricted, the increased mitochondria OCR can compensate by supplying NADPH to fuel NOX-derived ROS, and we demonstrate that this is supported by fatty acid oxidation. We have altered the text to further clarify these points on page 11, lines 10-15.

The reviewer noted, *“In addition, in all experiments authors did not look at any functional consequences of respiratory burst to link their findings with biological effect. As presented, Figure 2 provide very little information to support authors conclusion. The biological significance of the observations remains unclear.”*

We agree that establishing biological significance is important. Figure 2 was designed to identify a hereto undefined metabolic plasticity within oxidative subsets of neutrophils. The resultant effect of this novel mitochondrial arrangement was the production of ROS during limited glucose metabolism, an unreported phenomenon in the neutrophil field. To add biological context to these findings, we have demonstrated the ability of this population to

function in low glucose (Figure 3) including mounting a response to microbes such as zymosan (Figure S3a), added new data showing their ability to maintain ROS production at the tumor site (Fig 6c) and we show they can suppress T-cells in limiting glucose (figure 6g-i). Additionally, we have now added data showing that CRISPR-Cas9 deletion of *Kitl* in the tumor reduces the recruitment and ability of these neutrophils to perform respiratory burst *in situ* within the tumor and reduces tumor growth *in vivo* (new Fig6.C). Together these data show the functional significance of our findings and we have discussed this fact on page 18 (line 14) of the revised manuscript.

The reviewer noted, *“In Figures 4-6 authors repeatedly stated their conclusion about the role of glucose deprivation as important functional distinction between neutrophils in naïve and tumor-bearing mice. However, they provide no evidence that such disparity in glucose level exist in bone marrow and spleen between naïve and tumor-bearing mice.”*

We opted to use splenic neutrophils as an accessible peripheral neutrophil population. In doing so, we may have given the impression that we think that splenic neutrophils are adapted to an altered splenic niche in tumor bearing mice. This is not the case. We do not suggest that neutrophils in 4T1-bearing mice are adapted for the particular environment in spleen, blood or bone marrow when tumor is present, but rather are representative of systemic metabolic reprogramming that is exploited by the tumor at the tumor site. We did not intend to give the impression that the loss of mitochondrial function during maturity in naïve mice was an adaptation to high glucose, instead we and others would suggest that this is a correlative of the normal neutrophil maturation process (Maianski et al., 2004), perhaps contributing to their short life span. Therefore, what is happening here is the maintenance of an immature metabolic phenotype, regardless of niche requirements, that becomes advantageous at the tumor site, which is known to have low levels of glucose (Chang et al., 2015). To clarify this issue, we have inserted graphs demonstrating systemic reprogramming of mitochondrial function in blood-derived neutrophils between naïve and tumor bearing mice (new supplemental figure 4d) and altered our explanation of this in the text on page 14 of the revised manuscript.

The reviewer notes, *“It is easy to envision that their concept could work in tumor site where there is evidence of competition for glucose. However, they did not perform any direct experiments to assess function of neutrophils in tumors.”*

The reviewer is correct to point out that *in situ* neutrophil function would strengthen our conclusions. Unfortunately, the purification of large numbers of healthy neutrophils from 4T1 tumors has proved particularly technically difficult. Measurement of mitochondrial and glycolytic function in neutrophils retrieved from the tumor site repeatedly yield unreliable metabolic flux data (see reviewers figure 2a and b for examples). We therefore used splenic neutrophils as proxy population of peripheral neutrophils in many of our studies. Since our submission we have explored various alternative approaches to address the reviewer's concern. In the revised manuscript we have now employed an intravital luminol method for detection of ROS generation *in vivo* at the tumor site (new Fig 6a) discussed on page 18 (line

6). Others have shown this method to be dependent on neutrophil myeloperoxidase (MPO) activity (Alshetaiwi et al., 2013). Furthermore, we now demonstrate that upon dissociation of the tumor the Ly6G⁺ population is highly positive for ROS directly *ex vivo* (New Fig 6b and discussion on page 18, line 10). These exciting new data further support our contention that neutrophils are the source of this ROS *in vivo*. Furthermore, as a result of new experiments to address reviewer concerns, we have assessed the production of ROS by neutrophils in 4T1 tumors where tumor initiated kitl-cKit signaling has been ablated by genetically silencing *kitl* in the tumor. The absence of *kitl* production by the tumors led to a substantial reduction in the percentage of neutrophils producing ROS *ex vivo* (new Fig 6.c discussed on page 18, line 14 and page 25, line 9 of the revised manuscript. We thank the reviewer for pressing this issue, as the resulting data have substantially strengthened our conclusions.

As was Reviewer 1, Reviewer 2 was concerned about the veracity of the proof for our contention that c-Kit supports the phenotype we described here. The reviewer noted, *“c-kit inhibition is interesting. However, it is difficult to interpret these data the way suggested by the authors. C-kit is critically important for myelopoiesis, especially granulocytes. Inhibition of c-kit in tumor-bearing mice will dramatically reduce neutrophil population. Authors observed it in their study. It is difficult directly connect inhibition of c-kit with its effect on OCR. This may reflect selection “survival of the fittest” phenomenon.*

As noted in the response to Reviewer 1, comment 3 above, the reviewer is correct that c-kit antibodies would target haemopoietic progenitors in the bone marrow and although we do not see reversion of neutrophil number levels found in naïve mice, we do see a significant reduction in the total number of neutrophils. Therefore, to more directly assess possible crosstalk between the tumor and the neutrophil compartment we have now generated a SCF (Kitl) null tumor using CRISPR-Cas9 technology to specifically only target Kitl/c-Kit signaling initiated by the tumor, leaving normal hematopoiesis-associated Kitl signaling intact. These new data demonstrate that Kitl expression by the tumor is required for effective tumor growth *in vivo* (new Figure 5h), increased spleen sizes (new Fig 5i), increased neutrophil numbers (new fig 5j), and expanded c-Kit expression (new fig 5k). Most importantly, silencing of Kitl expression in the 4T1 tumor reduced, 1) neutrophil ATP synthase-dependent OCR and fatty acid oxidation (new fig 5m), 2) the ability to of these neutrophils to maintain *in vitro* respiratory burst when glucose utilization is limited (new fig 5n), and 3) tumor-associated neutrophil ROS production *ex vivo* (new Fig.6c). Taken together, these data demonstrate the importance of Kitl/c-Kit crosstalk in maintaining neutrophil mitochondrial metabolism in the tumor setting. These new data are discussed on pages 17, 18 and 25 of the revised manuscript.

The reviewer points out that, *“There is evidence that tumor associated MDSC are more suppressive per cell basis than spleen MDSC from the same mice. Some references include: Maenhout, S.K., et al. (2014) Int J Cancer 134, 1077-1090. Haverkamp, J.M., et al. (2011) Eur J Immunol 41, 749-759. Cimen Bozkus, C., et al. (2015) J Immunol 195, 5237-5250.”* and suggests that, *“...how this would fit to proposed concept is not clear.”*

Indeed, MDSC from the tumor site have been shown to be more suppressive than their splenic counterparts. Although directly defining this phenomenon was not a primary goal of our work, the new data we have added in response to reviewers comments show that ROS is produced by tumor-associated neutrophils directly *ex vivo*, whereas this is not the case in the spleen (new Fig. 6a, b). Moreover, this *ex vivo* ROS production is dependent on tumor-derived kitl along with the reversion of the oxidative phenotype we describe here (new Fig. 6c). These data, in addition to our finding that PMA stimulated, 4T1-elicited splenic neutrophils, exhibit increased suppressive capacity (Fig 6g) and can better maintain this suppression when glucose utilization is limited (Fig 6h,i), strongly supports our hypothesis that systemic metabolic reprogramming via tumor-derived kitl supports neutrophil ROS-dependent suppression within the inflammatory tumor environment. We again thank the reviewer for their suggestions and references which we have added to strengthen our conclusions (page 25, line 2).

Lastly, the reviewer suggest that, *“Putting glucose deprivation issue aside, authors concept in Figure 8 suggest that c-kit drives more ROS production by neutrophils in tumor-bearing hosts that made them suppressive. However, authors did not present data supporting this concept.”*

Although we do not intend make the conclusion that c-Kit directly drives more ROS production, as detailed in multiple figures and highlighted in this response, with the addition of our new data we now clearly show that kitl:cKit does supports the maintenance of neutrophils with the mitochondrial function required for ROS production in the glucose depleted tumor microenvironment. For example, we show that blockade of c-Kit or inhibition of Kitl production by the tumor reduced the ability of neutrophils to produce ROS when glucose metabolism is limited (Fig 5.g and New Fig. 5n), and that kitl silencing reduced the production of ROS producing neutrophils in the tumor (New Fig 6c). These new findings are addressed in the text (page 25).

Reviewer Figure 2

References for Response

- Alshetaiwi, H.S., S. Balivada, T.B. Shrestha, M. Pyle, M.T. Basel, S.H. Bossmann, and D.L. Troyer. 2013. Luminol-based bioluminescence imaging of mouse mammary tumors. *J Photochem Photobiol B*. 127:223-228.
- Chang, C.H., J. Qiu, D. O'Sullivan, M.D. Buck, T. Noguchi, J.D. Curtis, Q. Chen, M. Gindin, M.M. Gubin, G.J. van der Windt, E. Tonc, R.D. Schreiber, E.J. Pearce, and E.L. Pearce. 2015. Metabolic Competition in the Tumor Microenvironment Is a Driver of Cancer Progression. *Cell*. 162:1229-1241.
- Coffelt, S.B., K. Kersten, C.W. Doornebal, J. Weiden, K. Vrijland, C.S. Hau, N.J.M. Verstegen, M. Ciampricotti, L. Hawinkels, J. Jonkers, and K.E. de Visser. 2015. IL-17-producing gammadelta T cells and neutrophils conspire to promote breast cancer metastasis. *Nature*. 522:345-348.
- Coffelt, S.B., M.D. Wellenstein, and K.E. de Visser. 2016. Neutrophils in cancer: neutral no more. *Nat Rev Cancer*. 16:431-446.
- Davies, L.C., C.M. Rice, E.M. Palmieri, P.R. Taylor, D.B. Kuhns, and D.W. McVicar. 2017. Peritoneal tissue-resident macrophages are metabolically poised to engage microbes using tissue-niche fuels. *Nat Commun*. 8:2074.
- Delano, M.J., K.M. Kelly-Scumpia, T.C. Thayer, R.D. Winfield, P.O. Scumpia, A.G. Cuenca, P.B. Harrington, K.A. O'Malley, E. Warner, S. Gabrilovich, C.E. Mathews, D. Laface, P.G. Heyworth, R. Ramphal, R.M. Strieter, L.L. Moldawer, and P.A. Efron. 2011. Neutrophil mobilization from the bone marrow during polymicrobial sepsis is dependent on CXCL12 signaling. *J Immunol*. 187:911-918.
- Deniset, J.F., B.G. Surewaard, W.Y. Lee, and P. Kubes. 2017. Splenic Ly6G(high) mature and Ly6G(int) immature neutrophils contribute to eradication of *S. pneumoniae*. *J Exp Med*. 214:1333-1350.
- Kuonen, F., J. Laurent, C. Secondini, G. Lorusso, J.C. Stehle, T. Rausch, E. Faes-Van't Hull, G. Bieler, G.C. Alghisi, R. Schwendener, S. Andrejevic-Blant, R.O. Mirimanoff, and C. Rugg. 2012. Inhibition of the Kit ligand/c-Kit axis attenuates metastasis in a mouse model mimicking local breast cancer relapse after radiotherapy. *Clin Cancer Res*. 18:4365-4374.
- Maianski, N.A., J. Geissler, S.M. Srinivasula, E.S. Alnemri, D. Roos, and T.W. Kuijpers. 2004. Functional characterization of mitochondria in neutrophils: a role restricted to apoptosis. *Cell Death Differ*. 11:143-153.
- Pillay, J., T. Tak, V.M. Kamp, and L. Koenderman. 2013. Immune suppression by neutrophils and granulocytic myeloid-derived suppressor cells: similarities and differences. *Cell Mol Life Sci*. 70:3813-3827.
- Stejskal, S., I. Koutna, P. Matula, Z. Rucka, O. Danek, M. Maska, and M. Kozubek. 2010. The role of chromatin condensation during granulopoiesis in the regulation of gene cluster expression. *Epigenetics*. 5:758-766.
- Weisel, K.C., F. Bautz, G. Seitz, S. Yildirim, L. Kanz, and R. Mohle. 2009. Modulation of CXC chemokine receptor expression and function in human neutrophils during aging in vitro suggests a role in their clearance from circulation. *Mediators Inflamm*. 2009:790174.

REVIEWERS' COMMENTS:

Reviewer #1 (Remarks to the Author):

In the revised manuscript "Tumour-elicited neutrophils engage mitochondrial metabolism to circumvent nutrient limitations and maintain immune inhibition" by Rice et al., the authors have done an exemplary job in addressing the concerns noted from the initial review. The new experiments and data have significantly strengthened the story, particularly with respect to the explicit delineation of the cellular phenotyping and the elegant studies with ckit ligand -deficient tumors. This study is an important advancement for the fields of both tumor biology and neutrophil biology and represents a distinctive achievement in the field of neutrophil immunometabolism (heretofore markedly understudied). This work will certainly be of significant impact and interest to journal's readership.

Reviewer #2 (Remarks to the Author):

Authors made a concerted effort to address all my concerns. MS is substantially improved. I don't have any issues precluding me recommending this MS for publication.